# A cell type-aware framework for nominating non-coding variants in Mendelian regulatory disorders

Arthur S. Lee [1,2,3,4] ✉, Lauren J. Ayers [1], Michael Kosicki[5], Wai-Man Chan [1,6], Lydia N. Fozo [1], Brandon M. Pratt[1], Thomas E. Collins[1], Boxun Zhao [3,4,7,8], Matthew F. Rose[1,2,4,9,10,11], Alba Sanchis-Juan[4,12], Jack M. Fu [4,12,13], Isaac Wong [4,12], Xuefang Zhao[4,12,13], Alan P. Tenney[1,2,4], Cassia Lee[1,14], Kristen M. Laricchia[4], Brenda J. Barry[1,6], Victoria R. Bradford[1], Julie A. Jurgens[1,2,4], Eleina M. England[4], Monkol Lek [4], Daniel G. MacArthur[4,15,16], Eunjung Alice Lee[3,4,7,8], Michael E. Talkowski [4,12,13], Harrison Brand[4,12,13,17], Len A. Pennacchio [5] & Elizabeth C. Engle [1,2,3,4,6,7,11,18] ✉

Unsolved Mendelian cases often lack obvious pathogenic coding variants, suggesting potential non-coding etiologies. Here, we present a single cell multi-omic framework integrating embryonic mouse chromatin accessibility, histone modification, and gene expression assays to discover cranial motor neuron (cMN) *cis*-regulatory elements and subsequently nominate candidate non-coding variants in the congenital cranial dysinnervation disorders (CCDDs), a set of Mendelian disorders altering cMN development. We generate single cell epigenomic profiles for ~86,000 cMNs and related cell types, identifying ~250,000 accessible regulatory elements with cognate gene predictions for ~145,000 putative enhancers. We evaluate enhancer activity for 59 elements using an in vivo transgenic assay and validate 44 (75%), demonstrating that single cell accessibility can be a strong predictor of enhancer activity. Applying our cMN atlas to 899 whole genome sequences from 270 genetically unsolved CCDD pedigrees, we achieve significant reduction in our variant search space and nominate candidate variants predicted to regulate known CCDD disease genes *MAFB, PHOX2A, CHN1,* and *EBF3* – as well as candidates in recurrently mutated enhancers through peak- and gene-centric allelic aggregation. This work delivers non-coding variant discoveries of relevance to CCDDs and a generalizable framework for nominating non-coding variants of potentially high functional impact in other Mendelian disorders.

While the great majority of genetic variants associated with complex disease are common in the population and localize to non-coding sequences, less than 5% of the known Mendelian phenotype entries in OMIM have been attributed to non-coding mutations[1–4]. However, it remains unsettled the extent to which this disparity in coding:non-coding causal Mendelian variants is explained by the relative effect sizes of coding vs. non-coding variation, difficulty in deciphering the functional impact of non-coding variation, and/or ascertainment due to greater number and size of exome- versus genome-sequenced disease cohorts[1,5–8]. Nominating pathogenic non-coding variants in

Mendelian disease remains a major challenge due to a vastly increased search space (98% of the genome) relative to coding variants. Compounding this challenge is the lack of a generalizable rubric for nominating non-coding pathogenic variants relative to the more readily interpretable molecular and biochemical constraints governing protein-coding variant effects.

In recognition of these challenges, large-scale functional genomics projects such as ENCODE and Roadmap Epigenomics have provided valuable and expansive genome-wide functional information across a growing array of potentially disease-relevant tissues and cell types[9,10]. Such efforts reveal that the non-coding genome is abundant with *cis*-regulatory elements (cREs) - segments of non-coding DNA that regulate gene expression through transcription factor binding and three-dimensional physical interactions with their cognate genes. Biologically active cREs are associated with accessible chromatin, and combinations of accessible cREs vary dramatically among different cell types[11]. Therefore, understanding the chromatin accessibility landscape of cell types affected by disease is critical to identifying and interpreting disease-causing variation in the non-coding genome.

Disease-relevant developmental processes are disproportionately driven by regulation of gene expression[12,13], making congenital genetic disorders attractive candidates for non-coding etiologies. However, sampling developing human cell types remains particularly challenging, as samples are often restricted by cell location, assayable cells, invasiveness of sampling, and/or narrow windows of biologically relevant regulation of gene expression and development[14]. Thus, while fetal epigenomic reference sets are emerging for humans, samples are generally assayed at the whole-organ/tissue level and/or at later stages of development, making appropriate sampling and identification of early-born and rare cell types difficult[15]. By contrast, sample collection and marker-based enrichment in model organisms can achieve substantial representation of disease-relevant cell types at early stages of development[16–18].

The congenital cranial dysinnervation disorders (CCDDs) are Mendelian disorders in which movement of extraocular and/or cranial musculature are limited secondary to errors in the development of cranial motor neurons (cMNs) or the growth and guidance of their axons (Fig. 1a). Although a known subset of the CCDDs are caused by Mendelian protein-coding variants[19–28], a substantial proportion of cases remain unsolved by whole exome sequencing, including pedigrees with Mendelian inheritance patterns and cases with classic phenotypic presentations lacking corresponding mutations in the expected genes (representing potential locus heterogeneity)[29]. Moreover, most CCDD cases are sporadic or segregated in small dominant families for which non-coding variant prioritization is challenging.

The CCDDs represent an attractive test case for dissecting cell type-specific disorders, as defects in specific cMN populations are highly stereotyped with predictable corresponding human phenotypes[30]. By contrast, many complexes and even some Mendelian diseases are not immediately attributable to an unambiguous, singular cell type of interest, making assaying appropriate cell types a major challenge[31–33]. Moreover, while sampling and identification of developing cMNs at disease-relevant timepoints is challenging in developing human embryos, cMN birth, migration, axon growth/guidance, and mature anatomy/nerve branches are exquisitely conserved between humans and mice[30] (Fig. 1a). Motor neuron reporter mice permit sample collection and marker-based enrichment of cMNs at these key stages of development. Importantly, we previously demonstrated that such mouse models helped to characterize non-coding pathogenic variants that alter gene expression in hereditary congenital facial paresis type 1 (HCFP1), a disorder of facial weakness secondary to facial nerve (cMN7) maldevelopment[34]. Here, to comprehensively discover the repertoire of cREs underlying proper cMN development, we have generated a chromatin accessibility atlas of developing mouse cMNs and adjacent cell types. We subsequently use this atlas to reduce our

candidate variant search space (i.e., the total number of eligible variants) and ultimately interpret and nominate non-coding variants among 270 unsolved CCDD pedigrees (Fig. 1b and Supplementary Table 1).

## Results

### Defining disease-relevant cREs in the developing cMNs

To discover disease-relevant cREs and ultimately reduce our non-coding search space for nominating candidate pathogenic CCDD variants, we generated a single-cell atlas of embryonic mouse cMN chromatin accessibility. Using transgenic mice expressing GFP under the *Isl1^{MN}*:GFP or *Hb9*:GFP motor neuron reporters[35,36] (Fig. 1bi), we performed fluorescence-assisted microdissection and FACS-based enrichment of GFP-positive primary mouse embryonic oculomotor (cMN3), trochlear (cMN4), abducens (cMN6), facial (cMN7), hypoglossal (cMN12), spinal motor neurons (sMNs), and surrounding GFP-negative non-motor neuron cells (-neg), followed by droplet-based single-cell ATAC-seq (scATAC). cMN birth and development occur continuously over a period of weeks in early human embryos and days in mice[34,37]. More specifically, birthdating studies show that cMN3, 4, and 7 overlap mouse/mouse-equivalent stages e9.25 through e12.0[34,38–41]. In addition, their axons have extended into the periphery and are forming nerve branches at timepoints overlapping e10.5 and e11.5[42–47]. Finally, for the known CCDD genes, mRNA expression and/or observed cellular defects typically overlap key developmental timepoints e10.5 and e11.5 in mice−both for cellular identity-related transcription factor[42,48–51] and axon guidance-related[22,52,53] variants. Therefore, we captured e10.5 and e11.5 embryonic timepoints for each cMN sample, reasoning that a major proportion of both relevant cellular birth and initial axonal growth and guidance would be represented at one or both of these ages[34,37]. At these stages, these cranial nuclei contain only hundreds (cMN3, 4, 6) to thousands (cMN7, 12) of motor neurons per nucleus, per embryo[43,52,53].

We generated scATAC data across 20 unique sample types (cMN3/4, 6, 7, 12, and sMN for GFP-positive and -negative cells, each at e10.5 and e11.5), nine with biological replicates and two with technical replicates for 32 samples in total and sequenced them to high coverage (mean coverage = 48,772 reads per cell). We included GFP-negative cells to reduce uncertainty in peak calling, further increase representation from rare cell types, and capture regional-specific cell types that could harbor elements conferring non-cell-autonomous effects on cMN development. To generate a high-quality set of non-coding elements, we performed stringent quality control (Supplementary Fig. 1a–h, Methods). Altogether, we generated high-quality single-cell accessibility profiles for 86,089 (49,708 GFP-positive and 36,381 GFP-negative) cells, in some cases achieving substantial oversampling of cranial motor neurons in the developing mouse embryo (up to 23-fold cellular coverage). Our final dataset revealed prominent signals of expected nucleosome banding, a high fraction of reads in peaks ($\bar{x}_{frip} = 0.66$), transcription start site enrichment, and strong concordance between biological replicates (Fig. 1c, Supplementary Fig. 1d–h, and Supplementary Data 1). In addition to evaluating per-sample and per-cell metrics, we estimated a decrease in global accessibility over developmental time, consistent with observations in other developing cell types ($\beta_{time} = 0.049$, $p$ value $< 1 \times 10^{-15}$, linear regression, Supplementary Note 1)[54,55].

We performed bulk ATAC on a subset of microdissected and fluorescence-activated cell sorting (FACS)-purified cMN samples to evaluate the concordance between bulk and single-cell peak representation. As expected, bulk and single-cell cMN ATAC peaks are highly correlated in their matching dissected cell types (Supplementary Fig. 2a, b). scATAC peaks were enriched for intronic/distal annotations (relative to exonic/promoter annotations, OR = 1.9, $p$ value $< 2.2 \times 10^{-16}$, Fisher's exact test) compared to bulk ATAC intronic/distal annotations, thus better-capturing regions that harbor the overwhelming majority

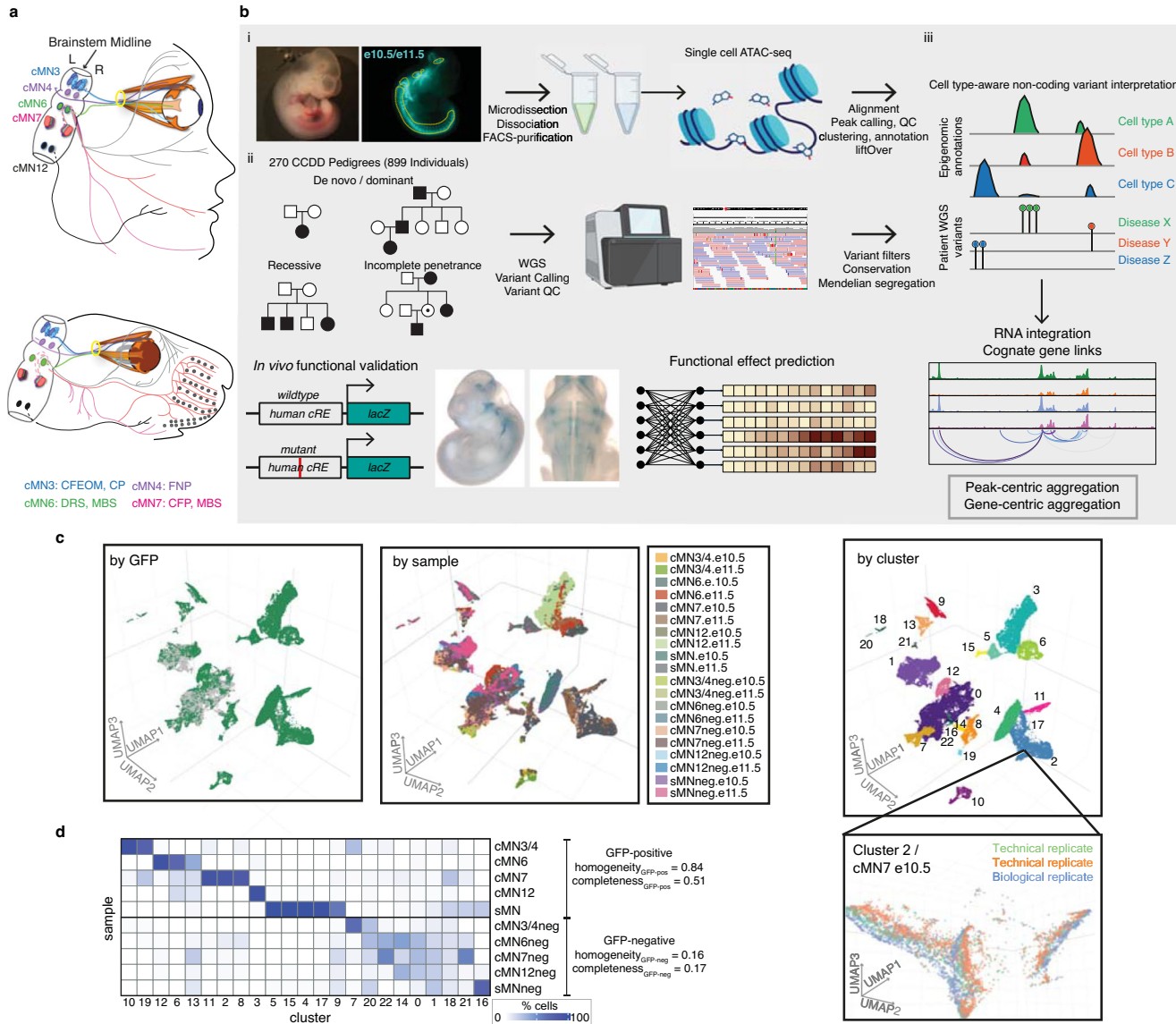

**Fig. 1 | Integrating Mendelian pedigrees with single-cell epigenomic data.**
**a** Schematic depicting a subset of human (top) and mouse (bottom) cMNs and their targeted muscles. cMN3 (blue) = oculomotor nucleus which innervates the inferior rectus, medial rectus, superior rectus, inferior oblique, and levator palpebrae superior muscles; cMN4 (purple) = trochlear nucleus which innervates the superior oblique muscle; cMN6 (green) = abducens nucleus which innervates the lateral rectus muscle (bisected); cMN7 (pink) = facial nucleus which innervates muscles of facial expression; cMN12 (black) = hypoglossal nucleus which innervates tongue muscles. Corresponding CCDDs for each cMN are listed under the diagram and color coded. CFEOM congenital fibrosis of the extraocular muscles, CP congenital ptosis, FNP fourth nerve palsy. **b** Overview of the experimental and computational approach. (i) Generating cell type-specific chromatin accessibility profiles. Brightfield and fluorescent images of e10.5 *Isl1^MN*:GFP embryo (left) from which cMNs are microdissected (yellow dotted lines, dissociated, FACS-purified (middle), followed by scATAC and data processing. (ii) WGS of 270 CCDD pedigrees (left; 899 individuals; sporadic and inherited cases) followed by joint variant

calling, QC, and Mendelian variant filtering (right). (iii) Integrating genome-wide non-coding variant calls with epigenomic annotations for variant nomination (top). To inform variant interpretation, we identify cognate genes (second row), aggregate candidate variants, generate functional variant effect predictions (third row), and validate top predictions in vivo (bottom). **c** UMAP embedding of single-cell chromatin accessibility profiles from 86,089 GFP-positive cMNs, sMNs, and their surrounding GFP-negative neuronal tissue colored by GFP reporter status (left, GFP-positive green, GFP-negative gray), sample (middle) and cluster (right). Gridlines in middle UMAP apply to left and right UMAPs as well. The inset shows the relative proximity of Cluster 2 cells dissected from the same cell type (cMN7 e10.5) from different technical and biological replicates. **d** Heatmap depicting proportions of dissected cells within each of the 23 major clusters. Homogeneity/completeness metrics are shown for GFP-positive versus -negative clusters. Figure 1b was created with BioRender.com and released under a Creative Commons Attribution-NonCommercial-NoDerivs 4.0 International license.

of regulatory elements (Supplementary Fig. 2c)[56]. Next, to test the cellular resolution of our scATAC data, we leveraged differences in the strategies used for bulk (cMN3 without cMN4) vs. scATAC dissection (cMN3 and cMN4 combined) and performed cluster analysis on cMN3/4 samples only (ad hoc clusters C1–C20, Supplementary Fig. 2a, d, e). We identified a significant overlap between ad hoc clusters C18 and C20 scATAC peaks with bulk cMN3 peaks. Moreover, we confirmed the

accessibility of known cMN3 markers in C18 and C20, and cMN4 markers in C19[57,58] (Supplementary Fig. 2e). When comparing the scATAC peaks to bulk ATAC peaks in ENCODE[9] sampled from major developing brain regions (forebrain, midbrain, hindbrain) at comparable timepoints, we observed diminished overlap for GFP-positive cMN samples relative to GFP-negative samples (Supplementary Fig. 3a). Further stratifying scATAC peaks based on cell type specificity

scores[59] revealed that highly specific scATAC peaks had consistently lower bulk coverage than peaks with low specificity (Supplementary Fig. 3b, c), in keeping with findings that cell-type specific regulatory elements often act within small populations of cells and may be more difficult to capture and annotate with bulk methods[60,61].

To further distinguish between rare, distinct cell types, we adopted an iterative clustering strategy (Methods)[59]. We first identified 23 major clusters that correspond with ground truth dissected cell types based on known anatomy (Fig. 1c, d and Supplementary Data 2). Overall, GFP-positive clusters demonstrated much more uniform sample membership than GFP-negative clusters, as reflected by their differences in cluster homogeneity[62] ($h_{\text{gfp-positive}} = 0.84$ vs. $h_{\text{gfp-negative}} = 0.16$) and purity metrics (Fig. 1d, Supplementary Fig. 4a, and Supplementary Table 2, Methods). Upon examining differentially accessible genes and elements through manual curation, review of the literature, and gene ontology analysis, we assigned provisional cell identities to the 23 major clusters, of which ten clusters are cranial, and five are spinal motor neurons based on dissection origin, and nine are cranial, and four are spinal motor neurons based on putative annotation (Supplementary Data 2). To further resolve the heterogeneity within clusters and to identify functionally and anatomically coherent subpopulations, we performed iterative clustering[59] on each major cluster and identified 132 unique subclusters (Supplementary Fig. 4bi, ii). Of these, 59 have GFP-positive membership >90%, representing highly pure motor neuron populations (Supplementary Fig. 4c). We observe even more distinct anatomic/temporal membership at the subcluster level, particularly for GFP-negative samples (subcluster homogeneity $h_{\text{gfp-positive}} = 0.87$ vs. $h_{\text{gfp-negative}} = 0.43$). These findings are consistent with highly dynamic and proliferative neurodevelopmental processes during this time period[12]. Neither major cluster nor subcluster membership was well-explained by the experimental batch (Supplementary Fig. 4d, Methods).

## cMN cRE functional conservation between mouse and human

Common disease risk loci tend to overlap non-coding accessible chromatin in their corresponding cell types - including accessible chromatin that is more readily ascertained in mouse versus human tissues[15,59]. However, with the exception of a few exemplary elements (e.g., refs. [63–65]), the extent of overlap between human/mouse elements underlying Mendelian traits is largely unknown. Therefore, to evaluate the functional conservation of cREs in our cranial motor neuron atlas, we performed in vivo humanized enhancer assays on a curated subset ($n = 26$) of our candidate scATAC peaks ($n = 255,804$ total) that were absent from the VISTA enhancer database ($n = 3384$)[66] and had peak accessibility/specificity in cMNs ($n = 45,813$)[59] and general signatures of enhancer function (i.e., evolutionary conservation and non-cMN-specific histone modification data[67], Supplementary Table 3, Methods). Importantly, these peaks and features were not randomly selected and, therefore, do not necessarily reflect overall patterns across the genome (see refs. [68,69]). We detected positive enhancer activity (any reporter expression) in 65% (17/26) of candidates. Moreover, 11 of the 17 validated enhancers (65%, 42% overall) recapitulate the anatomic expression patterns (motor neuron expression) predicted from the scATAC accessibility profiles to the resolution of individual nuclei/nerves. For these curated examples, we find that high-quality single-cell accessibility profiles are highly predictive of cell type-specific regulatory activity.

## Motif enrichment and footprinting reveal putative cMN regulators

To identify transcription factors/motifs responsible for cell type identity, we performed motif enrichment and aggregated footprinting analysis across all 23 major clusters and identified lineage-specific cMN transcription factor/motif relationships (Fig. 2a, b). For example, we identified significant motif and footprinting enrichment of midbrain

transcription factor OTX1 in populations corresponding to developing oculomotor/trochlear motor neurons (cluster cMN3/4.10) and the midbrain-hindbrain boundary (cluster MHB.7)[70]. We also identified notable footprints for ONECUT2 in multiple motor neuron populations, including cMN3/4, cMN7, and putative pre-enteric neural crest-derived cells (clusters cMN3/4.19, cMN7.11, enteric.17; Fig. 2b). Importantly, we detected positive footprint signals for known lineage-specific regulators such as JunD footprints in the spinal and lymphoid lineages[71,72] (clusters sMN.15, WBC.18) and GATA1 footprints in the erythroid lineage[73] (cluster RBC.20; Fig. 2b). Due to the relatively high homogeneity across the motor neuron clusters, we also compared motif enrichment across broader anatomic/functional classes of motor neurons and brain regions (Fig. 2c). We identified strong enrichment of regional markers such as DMBX1[74] in midbrain samples (i.e., cMN3/4 and cMN3/4neg). We also found motifs enriched among the ocular motor neurons (i.e., cMN3/4 and cMN6) such as PAX5, providing potential avenues for comparative studies.

## Assigning cell type-specific cREs to their cognate genes

A chief barrier to interpreting non-coding regulatory elements is identifying their *cis*-target genes. While enhancers often regulate adjacent genes, many important regulatory links also occur over much longer distances, including known disease-causing events[63,65,75–79]. Therefore, we generated scRNA data from GFP-positive and -negative cMN3/4, 6, and 7 at e10.5 and e11.5 (Methods) using reporter constructs, microdissection, and collection strategies analogous to those used to generate the scATAC datasets. We then integrated these scRNA data with the cMN chromatin accessibility data to generate peak-to-gene links connecting cREs to target genes at the single cell level for putative cREs within ±500 kb of a given gene (see Methods[80–82]). In total, we identified 145,073 known and putative enhancers with peak-to-gene links across the 23 clusters (median = 2 genes per enhancer, range = 1–37; Supplementary Data 3).

Because the accuracy of peak-to-gene links inferred from separate assays of ATAC and RNA data (diagonal integration, see ref. [83]) depends heavily on cell pairings, we performed multiple analyses to ensure that both our ATAC-RNA pairings and gene expression estimates were well calibrated. We compared our imputed single-cell gene expression estimates to independently collected in-house bulk RNA-seq experiments from cMN3, 4, 6, and 7 at e10.5 and e11.5 annotated with ground truth dissection labels (Methods). We identified strong positive concordance between imputed gene expression and measured bulk RNAseq signal in the appropriate cell types (Fig. 3a, b). We also found that our ATAC-RNA pairings and peak-to-gene links were sensitive to the cellular composition of our scRNA integration data. If the identical master peakset was compared to published scRNA data from e10.5 to e11.5 mouse brain (MOCA neuro) or e9.5 to e13.5 mouse heart (MOCA cardiac)[84] in place of our cMN-enriched scRNA data, we found fewer significant peak-to-gene links and fewer concordant cognate genes (Fig. 3c–f; Methods).

Next, we performed a joint ATAC-RNA coassay (scMultiome) on a subset of e11.5 GFP-positive cells represented in our main scATAC dataset (cMN3/4, cMN7, cMN12, and sMN), thereby allowing us to benchmark our inferred ATAC-RNA pairings against direct experimental measurements (vertical integration; Supplementary Fig. 5a–e). We found that scMultiome peak-to-gene links were highly concordant with our original scATAC peak-to-gene links (Fig. 3g–i). We then examined the single-cell accessibility profiles of four highly characterized cMN enhancers with known connection to the *Isl1* gene—a cMN master regulator embedded in a gene desert (Fig. 4a–c)[66,85]. Strikingly, both by diagonal and vertical integration, we found that for these four enhancers (mm933, CREST1/hs1419, CREST3/hs215, and hs1321), chromatin accessibility alone was a significant predictor of in vivo *Isl1* expression patterns in the anatomically appropriate cMN (Fig. 4d, e and Supplementary Fig. 5d; Wald test $p$ value = 0.011; Methods).

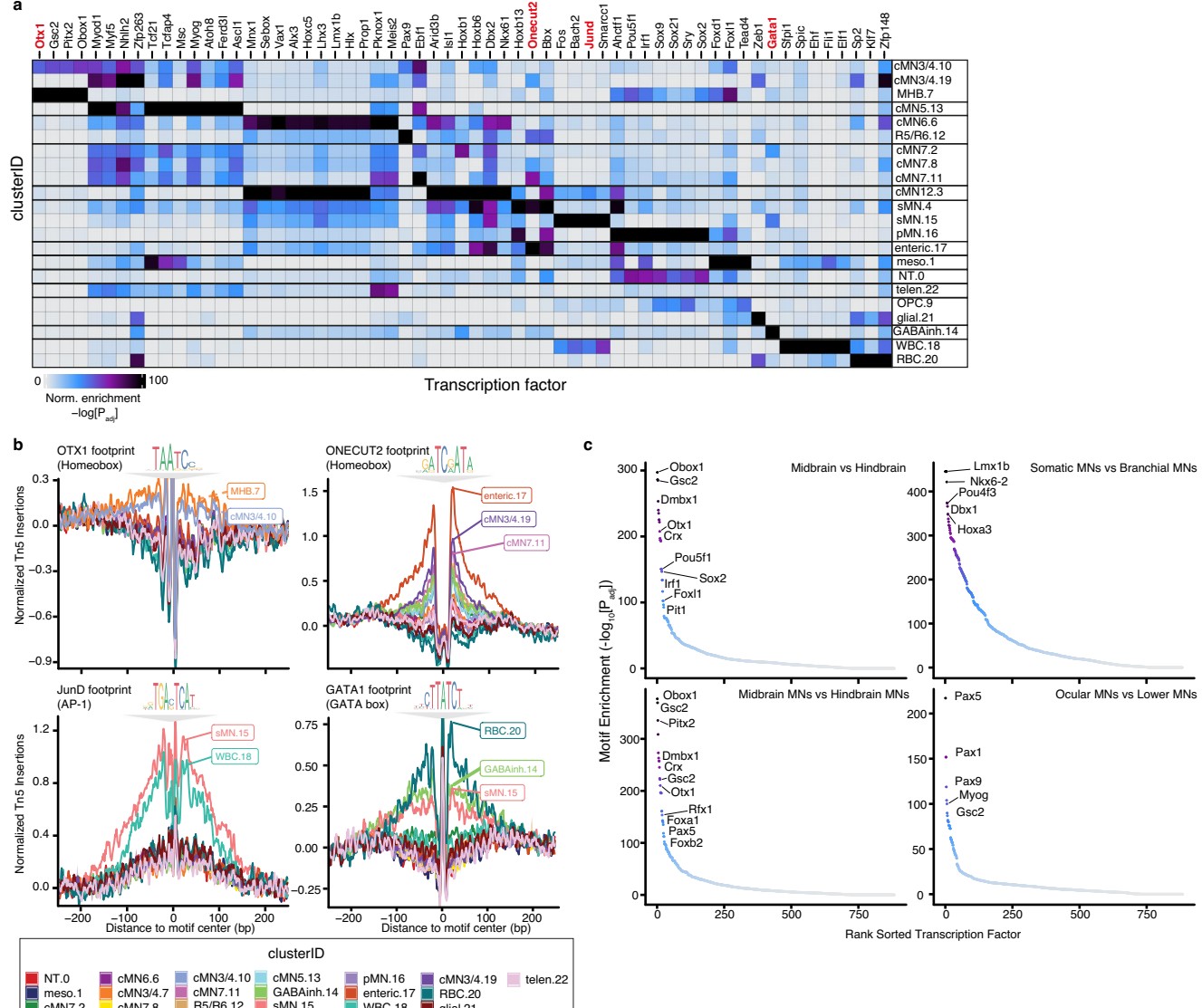

**Fig. 2 | Motif enrichment and aggregate footprint analyses distinguish cell type-specific TF binding motifs. a** Heatmap depicting enriched transcription factor binding motifs within differentially accessible peaks by cluster. Each entry is defined by its cluster identity (clusterID.clusterNumber). Corresponding cluster IDs and annotations are depicted. Color scale represents one-sided hypergeometric-enrichment *p* values adjusted for multiple testing for each cluster and motif. Specific motifs and motif families vary significantly amongst clusters. Cluster annotations are defined in Supplementary Data 2. **b** Aggregated subtraction-normalized footprinting profiles for a subset of cluster-enriched transcription factors (OTX1, ONECUT2, JunD, and GATA1) from (**a**), centered on their respective binding motifs. Specific clusters display positive evidence for TF

motif binding for each motif. Corresponding motif position weight matrices from the CIS-BP database are depicted above each profile. Cluster IDs with corresponding colors are below. **c** Motif enrichment comparing broad classes of neuronal subtypes. *P* values are from one-sided hypergeometric-enrichment testing adjusted for multiple testing. Midbrain subtype contains motifs from cMN3/4neg cells; hindbrain from cMN6neg, cMN7neg, and cMN12neg cells; somatic MN from cMN3/4, cMN6, and cMN12 GFP-positive cells; branchial MN are from cMN7 GFP-positive cells; midbrain MN are cMN3/4 GFP-positive cells; hindbrain MN are cMN6, cMN7, and cMN12 GFP-positive cells; ocular MN are cMN3/4 and cMN6 GFP-positive cells; lower MN are cMN7, cMN12, and sMN GFP-positive cells. For each graph, the first listed subtype is enriched relative to the second listed subtype.

Lastly, we integrated histone modification signatures into our enhancer predictions by performing H3K27Ac scCUT&Tag on e11.5 GFP-positive cMN3/4, cMN6, and cMN7 and e10.5 cMN7 (seven replicates total) and generated activity-by-contact (ABC) enhancer predictions for each cell type (Methods[86,87]). Of 6072 total ABC enhancers, 4925 (81%) directly overlapped our peak-to-gene links, including multiple in vivo ground truth enhancers (Supplementary Fig. 6a, Figs. 3i, 4a, and Supplementary Data 4). Because the availability of cell type-specific experimental data can be a limiting factor in accurate enhancer prediction, we assessed the relative contribution of cell type-specific chromatin accessibility versus histone modification data to ABC prediction accuracy. Specifically, among 67

annotated cMN enhancers in the VISTA enhancer database (visualized at e11.5 by the presence of beta-galactosidase in the nucleus and/or nerve), 49 had some evidence of expression in cranial nerve (CN)7. Among these, we identified seven that had both visible CN7 expression and ABC cMN7 enhancer predictions at e11.5. For all seven enhancers (100%), ABC cognate gene predictions were concordant with peak-to-gene predictions. We then reran our ABC predictions, replacing either our cMN7 ATAC data with mouse embryonic limb e11.5 ATAC data (ENCODE ENCSR377YDY; Limb ATAC) or our cMN7 histone modification data with mouse limb histone modification data (ENCODE ENCSR897WBY; Limb H3K27Ac) and compared predictions. Substituting limb ATAC for cMN7 ATAC data resulted in only

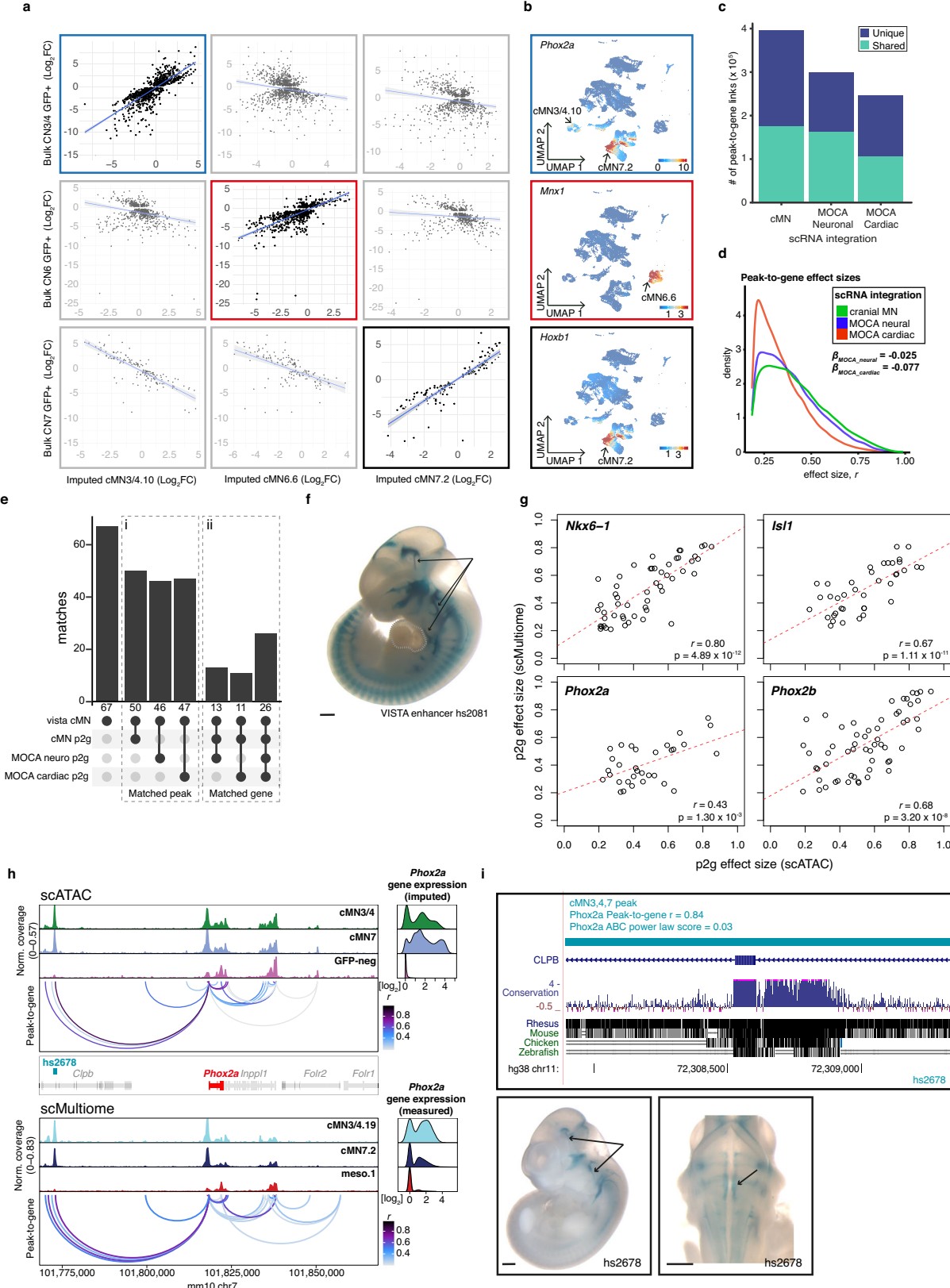

14% (1/7) concordance, while substituting limb H3K27Ac for cMN7 H3K27Ac data resulted in 57% (4/7) concordance (Supplementary Fig. 6b). Thus, for this curated set of data, we find that cell type-specific ATAC signal is a better predictor of reproducible cognate gene predictions than cell type-specific histone modification signal or non-cell-type-specific ATAC signal.

**Embryonic mouse chromatin accessibility atlas**

In summary, we generated a chromatin accessibility atlas of the developing cMNs and surrounding cell types. We combined GFP-positive ($n = 49,708$) and -negative ($n = 36,381$) cells to improve joint peak calling performance and to capture potential regional heterogeneity of non-motor neuron cell types as well as motor neuron

**Fig. 3 | Effects of RNA input data on peak-to-gene accuracy. a** Imputed gene expression values projected onto scATAC clusters cMN3/4.10, cMN6.6, and cMN7.2 versus measured gene expression values from bulk RNA-seq samples. Error bands represent 95% confidence intervals for each fitted model. **b** Feature plots depicting imputed gene expression for cMN marker genes *Phox2a, Mnx1*, and *Hoxb1*. **c** Total number of unique and shared peak-to-gene links using different scRNA integration datasets (cMN, MOCA Neuronal, MOCA Cardiac) against the common scATAC cMN peakset. **d** Distribution of peak-to-gene effect sizes using different scRNA integration datasets. Estimated effect sizes are significantly stronger for cMN scRNA integration relative to MOCA neuro ($\beta_{MOCA\_neuro} = -0.077$, $p < 2 \times 10^{-16}$, linear regression) and cardiac ($\beta_{MOCA\_cardiac} = -0.025$, $p < 2 \times 10^{-16}$, linear regression) integrations. **e** Barplot depicting peak-to-gene elements from different scRNA integrations overlapping experimentally validated cMN enhancers (vista cMN). (i) Matched peak indicates intersect overlapping peaks irrespective of predicted cognate gene. (ii) The matched gene indicates distinct overlapping peaks and identical cognate genes. **f** In vivo enhancer assay for VISTA enhancer hs2081 ($n = 4$

embryos) overlapping a predicted peak-to-gene link using cMN versus MOCA cardiac scRNA input. Enhancer activity is positive in cranial nerves 3, 7, and 12 (arrows); negative in heart (dotted lines). **g** Comparing scATAC versus scMultiome peak-to-gene effect sizes for marker genes *Nkx6-1, Isl1, Phox2a*, and *Phox2b*. Linear regression coefficients and nominal *p* values are shown. **h** scATAC and scMultiome accessibility profiles with peak-to-gene connections for a 100 kb window centered around *Phox2a*. hs2678 ($n = 5$ embryos) is accessible in cMN3/4 and cMN7 and is predicted to enhance *Phox2a* by scATAC ($r = 0.84$) and scMultiome ($r = 0.69$). **i** (Top) hs2678 is 70.3 kb distal to human *PHOX2A* and is embedded in the coding and intronic sequence of *CLPB*. (Bottom) In vivo enhancer assay using human hs2678 ($n = 5$ embryos) sequence is positive in cMN3 and cMN7 (arrows). Reporter expression views are shown as lateral (left) and dorsal through the fourth ventricle (right). Scale bars in **f** and **i** = 500 μm and are approximate measurements based on E11.5 embryo average crown-rump length of 6 mm. Source data are provided as a Source Data file.

progenitors[88]. Cluster analysis revealed nine putative cMN, four putative sMN, and multiple non-MN/non-neuronal clusters (of 23 total). Although sMNs are not directly implicated in CCDDs, they may provide value for comparative studies with cMNs[89,90]. We also performed iterative clustering to identify 132 subclusters, of which 58 are highly pure groups of motor neurons. Although we are currently unable to annotate subclusters, more detailed spatial and developmental profiling of the cMN subnuclei may help to identify functionally relevant groups of cells and/or cell states. Finally, a high quality and cell type-specific catalog of cMN elements and their cognate genes can be used to interpret and prioritize CCDD variants, as we describe below.

## Human phenotypes and genome sequencing

We enrolled and phenotyped 899 individuals (356 affected, 543 family members) across 270 pedigrees with CCDDs. About 202 probands were sporadic (simplex) cases enrolled as trios, while 42 and 19 pedigrees displayed clear dominant or recessive inheritance patterns, respectively (Supplementary Data 5). Of note, the dominant pedigrees included three with congenital facial weakness that we have reported to harbor pathogenic SNVs in a non-coding peak, cRE2, within the HCFP1 locus on chromosome 3[34]. The CCDDs included congenital fibrosis of the extraocular muscles (CFEOM), congenital ptosis (CP), Marcus Gunn jaw winking (MGJW), fourth nerve palsy (FNP), Duane retraction syndrome (DRS), congenital facial palsy (CFP), and Moebius syndrome (MBS) (Supplementary Data 5). Importantly, these CCDD phenotypes can be connected to the maldevelopment of their disease-relevant cMNs: CFEOM to cMN3/4, CP to the superior branch of cMN3, FNP to cMN4, DRS to cMN6, CFP to cMN7, and MBS to cMNs 6 and 7 (Fig. 1a and Supplementary Table 1).

We performed whole genome sequencing (WGS) and variant calling of the 899 individuals (Methods). First, to generate a comprehensive and unbiased set of genetically plausible candidates, we performed joint single nucleotide variant (SNV) and insertion/deletion (indel) genotyping, quality control, and variant frequency estimation from >15,000 WGS reference samples in the Genome Aggregation Database (gnomAD)[91,92]. We identified 54,804,014 SNV/indels across the cohort. Of these, 1,150,021 (2.1%) were annotated as exonic, 18,761,202 (34.2%) intronic, 34,512,518 (63.0%) intergenic, and 364,300 (0.7%) within promoters. We next performed initial SNV/indel variant filtering based on established and custom criteria, including genotype quality, allele frequency, and conservation (Methods)[93,94]. We incorporated family structures to include or exclude genetically plausible candidates that are consistent with known modes of Mendelian inheritance. Applying this approach to the 54,804,014 SNVs/indels across our cohort, we identified 26,000 plausible candidates (mean = 101 variants per pedigree). We also performed short-read structural variant (SV) discovery using an ensemble SV algorithm (GATK-SV) that was comparable to SVs generated in gnomAD and the 1000 Genomes

Project[91,95] and identified 221,857 total SVs (including transposable elements and other complex events). These WGS from deeply phenotyped CCDD pedigrees present a rich catalog of otherwise unannotated candidate Mendelian disease variants.

## Integrating epigenomic filters with human WGS variants

To further refine the 26,000 CCDD candidate SNVs/indels, we eliminated from further analysis 37 pedigrees definitively solved by coding variants and reported separately[96], and then applied cell type-specific filters from our scATAC peakset to each CCDD phenotype (Methods). We identified 5353 unique segregating SNVs/indels (3163 de novo/dominant, 1173 homozygous recessive, and 1017 compound heterozygous) that overlapped cMN-relevant peaks of accessible chromatin (23.6 and 13.6 candidates per monoallelic and biallelic pedigree, respectively). We only considered compound heterozygous variants observed in the same peak. Applying an analogous cell type-aware framework for SVs, we identified 115 candidates (72 deletions, 27 duplications, 1 inversion, 13 mobile element insertions, and 2 complex rearrangements encompassing multiple classes of SVs). There was substantial overlap between candidate variants and CCDD-relevant cMN peaks when compared to size-matched randomized peaks (median de novo Z-score = 0.9, median dominant inherited Z-score = 30.1, *p* value <2.0 × 10⁻⁴, permutation test; Supplementary Table 4). Using these 5468 cell type-aware non-coding CCDD candidate SNVs/indels/SVs and ATAC-based cMN enhancers, we next identified strong candidate variants using peak-centric or gene-centric approaches that aggregate variants in the same peak or in peaks with a shared target gene, respectively (Supplementary Fig. 7).

We adopted a gene-centric aggregation approach by first identifying non-coding candidate variants connected to a restricted set of 17 genes with prior association with CCDDs[19,21–26,28,42,44,97–103]. We identified non-coding variants connected to four: *MAFB*, *PHOX2A*, *CHN1*, and *EBF3* (Table 1). We also identified compound heterozygous variants connected to *ISL1* in a proband with CFP; *ISL1* is not a known disease gene but is a master cMN regulator (Supplementary Fig. 8a, b). Extending this approach to the entire genome, we identified 559 genes with multiple connected peaks containing dominant candidate variants (multi-hit genes, range of connected variants per gene = 2–6, Supplementary Data 6).

*EBF3* is an example of both a CCDD gene and a multi-hit gene. Monoallelic *EBF3* loss-of-function (LoF) coding mutations cause Hypotonia, Ataxia, and Delayed Development Syndrome (HADDS)[104], and two individuals are reported with HADDS and DRS, one with a coding missense variant and one with a splice site variant[103,105]. We identified a series of coding and noncoding *EBF3* variants (Supplementary Data 7). Two probands with DRS have large de novo multi-gene deletions, and one proband with fourth nerve palsy has a de novo stop-gain coding variant (Fig. 5a, b). These three individuals also have

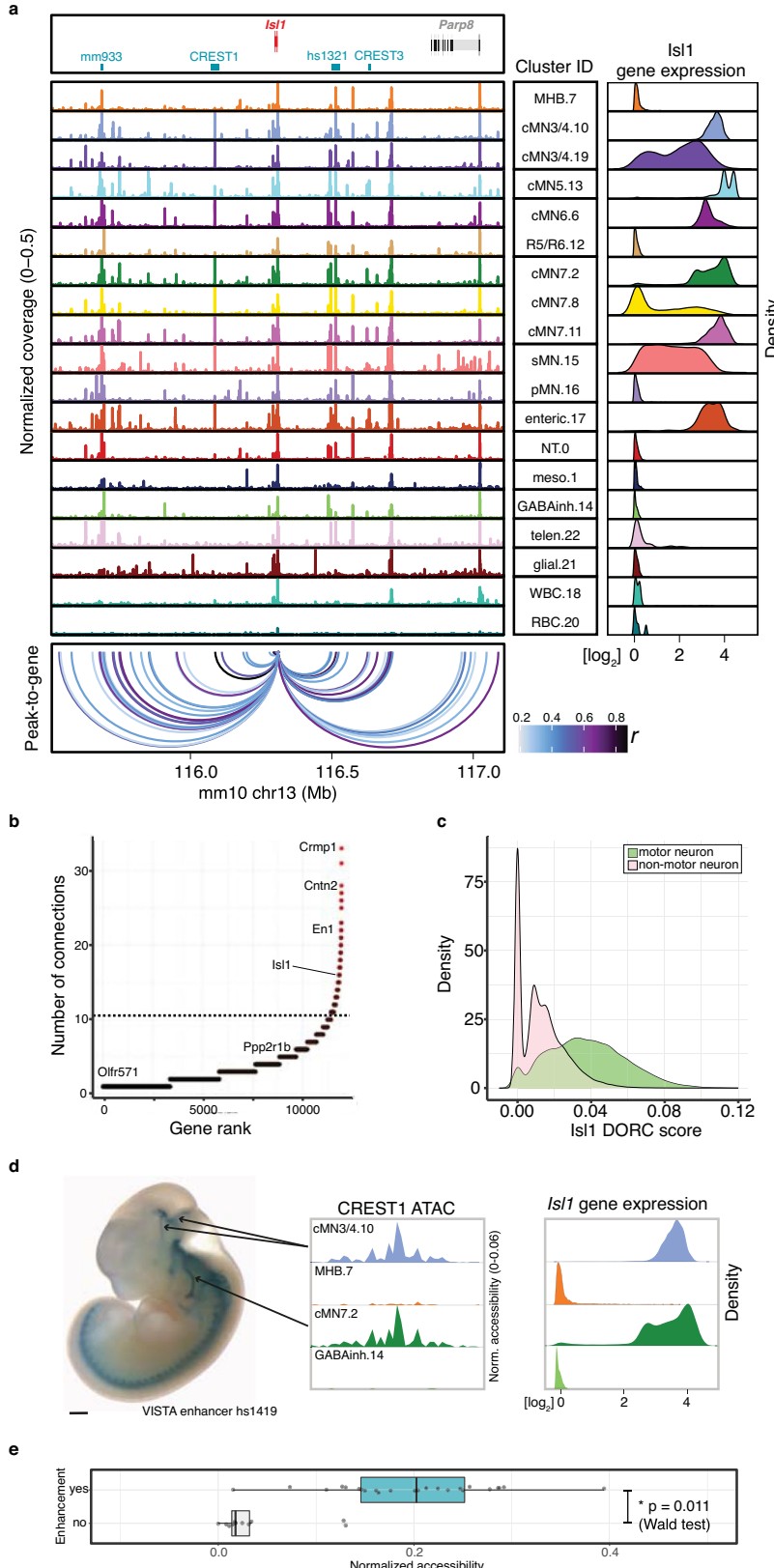

**Nature Communications** | (2024)15:8268

phenotypes consistent with HADDS. We also identified three inherited non-coding candidate variants with peak-to-gene connections to *EBF3* (Fig. 5b), which were cosegregated in an autosomal dominant fashion. Pedigrees S63 (distal indel), S176 (intronic SNV), and S95 (intronic SNV) segregate non-coding candidate variants with syndromic ptosis, isolated MGJW, and isolated ptosis, respectively. The multiple ocular CCDD phenotypes we observed potentially reflect pleiotropic consequences of *EBF3* variants, a phenomenon previously observed for coding mutations in other CCDD genes[106]. Moreover, the differences in syndromic versus isolated phenotypes may reflect more cell type-specific effects of non-coding variants. Indeed, multiple Mendelian disorders with non-coding etiologies are restricted to isolated cell

**Fig. 4 | Exceptional gene regulation of cranial motor neuron master regulator *Isl1*. a** Pseudobulked chromatin accessibility profiles for all annotated clusters over a 1.5 Mb window centered about *Isl1*. Imputed gene expression profiles for each cluster are shown to the right. *Isl1* is located within a gene desert with the nearest up- and downstream flanking genes 1.2 and 0.7 Mb away, respectively. Peak-to-gene predictions match known *Isl1* enhancers CREST1 and CREST3 (https://doi.org/10.1016/j.ydbio.2004.11.031); mm933 in multiple cranial motor nerves, dorsal root ganglion, and nose; hs1321 in multiple cranial motor nerves and forebrain) and identify additional putative enhancers surrounding *Isl1*. **b** The number of normalized regulatory connections for each rank-ordered gene. *Isl1* ranks in the top 1% of all genes with at least one regulatory connection. The inflection point of the plotted function is demarcated with a dotted line. **c** Per-cell domain of regulatory chromatin (DORC) scores for *Isl1* gene. DORC scores are significantly higher for cells from motor neuron clusters relative to non-motor neuron clusters (*p* value

$<1 \times 10^{-15}$, ANOVA). **d** (Left) Lateral whole mount In vivo reporter assay testing CREST1 (VISTA enhancer hs1419; *n* = 7 embryos) enhancer activity. CREST1 drives expression in cranial nerves 3, 4, and 7 (black lines). (Right) Single-cell ATAC profiles and imputed gene expression for a subset of corresponding clusters. CREST1 accessibility and *Isl1* gene expression are positively correlated with in vivo expression patterns. **e** Boxplot depicting normalized accessibility levels for enhancers CREST1, CREST3, mm933, and hs1321 within nine scATAC clusters corresponding to distinct anatomic regions. Manually scored enhancer activity is significantly correlated with normalized accessibility (*p* value = 0.011, Wald test, two-sided). Center line: median; box limits: upper and lower quartiles; whiskers – 1.5 × interquartile range. Scale bar in **d** = 500 μm and are approximate measurements based on E11.5 embryo average crown-rump length of 6 mm. Source data are provided as a Source Data file.

types or organ systems[65,75,107–110]. Notably, *EBF3* is broadly expressed across cMNs (Fig. 5c) and is one of the most constrained genes in the human genome as measured by depletion of coding LoF variants in gnomAD and SV dosage sensitivity (loeuf = 0.1500 and pHaplo = 0.9996, respectively; Fig. 5d)[92,111,112]. We observed exceptional conservation of non-coding elements within *EBF3* introns, comparable to or exceeding exonic conservation. This includes the ultraconserved element UCE318 (Fig. 5b, e) located in intron 6 with a peak-to-gene link to *EBF3* ($r = 0.69$, FDR = $6.2 \times 10^{-69}$). We also detected a peak-to-gene link from VISTA enhancer hs737 to *EBF3* ($r = 0.60$, FDR = $4.8 \times 10^{-49}$), an element located >1.2 Mb upstream of the gene that was previously reported to be linked to *EBF3* and to harbor de novo variants associated with autism with hypotonia and/or motor delay[113]. We did not observe any candidate variants in UCE318, consistent with depletion of both disease-causing and polymorphic variation within ultraconserved elements[114], nor in hs737, consistent with its non-CCDD phenotype.

Second, we took a peak-centric approach by examining all 5468 (5353 SNV/indels, 115 SVs) cell type aware non-coding variants, irrespective of cognate gene. When aggregating variants within disease-relevant cMN peaks, we identified 28 peaks harboring variants in more than one pedigree (multi-hit peaks). Fourteen multi-hit peaks contained variants obeying a dominant mode of inheritance (28 unique dominant/de novo variants with one variant present in two unrelated families, including the three pathogenic chromosome 3 cRE2 SNVs that cause CFP[34]), and 14 multi-hit peaks contained variants obeying a recessive mode of inheritance (35 unique recessive variants; Supplementary Data 8). Moreover, ten of these multi-hit peaks were also linked to multi-hit genes. Because enhancers confer cell type-specific function, we reasoned that true functional non-coding SNV/indels are less likely than coding variants to cause syndromic, multi-system birth defects. Interestingly, when stratifying pedigrees by isolated/syndromic status, we found a significant overrepresentation of isolated CCDD phenotypes for our dominant multi-hit peaks (OR = 5.9, *p* value = $2.3 \times 10^{-3}$, Fisher's exact test), but not for our recessive multi-hit peaks (OR = 0.8, *p* value = 0.64).

Among the multi-hit peaks, we identified 3.6 kb homozygous non-coding deletions centered over peak hs2757 in two probands with DRS (Table 1; one with segregation of sensorineural hearing loss as a dominant trait); in each case, the consanguineous parents were heterozygous for the deletion. The probands had extended runs of homozygosity with a shared 16 kb haplotype surrounding the deletion, consistent with a founder mutation (Fig. 6a–c). hs2757 is broadly accessible in multiple cMN populations, including cMN6, and is located 307 kb upstream of its nearest gene, *MN1*; *MN1* imputed gene expression estimates revealed widespread expression across all sampled cell types, including cMN6 (Fig. 6d)[100,115]. Monoallelic LoF coding mutations in *MN1* cause CEBALID syndrome, a disorder affecting multiple organ systems. A subset of individuals with coding variants in *MN1* are reported to have CEBALID syndrome with DRS[100]. *MN1* is constrained against LoF variation and dosage changes (loeuf = 0.087;

pHaplo = 0.9901, Fig. 6e)[92,111]. We performed in vivo enhancer testing on hs2757 which revealed reporter expression in a subset of tissues with known *Mn1* expression[115], including expression in the hindbrain overlapping the anatomic territory of cMN6 (Fig. 6f). Surprisingly, in this case, we did not observe a peak-to-gene link between hs2757 and *Mn1* and did observe links with genes *C130026L21Rik* (whose sequence maps to a different chromosome in human) and *Pitpnb* (Supplementary Data 8). Multiple scenarios may explain this result, such as active *Mn1* enhancement occurring prior to the mouse e10.5-e11.5 window investigated here. Alternatively, our regression-based peak-to-gene estimates may be less sensitive at detecting enhancers for ubiquitously expressed genes, a phenomenon previously observed for other enhancer prediction methods[86].

## Mechanistic insights of non-coding disease variants

Mendelian disease variant interpretation often relies on variant-level predictions of pathogenicity[116,117]. However, such prediction algorithms are typically agnostic to cell type- or disease-specific information. More recent approaches have incorporated cell type-specific epigenomic data to annotate non-coding variants in common diseases[61,118,119]. To leverage our cell type-specific accessibility profiles for variant-level functional interpretation, we trained a convolutional neural network[120] to generate cell type-specific predictions of chromatin accessibility for each cranial motor neuron population. When evaluating held-out test data, we consistently observed high concordance between our accessibility predictions and true scATAC coverage for each cell type (median Pearson's $r = 0.84$; range = 0.81 to 0.95; Fig. 7a and Supplementary Fig. 9a–c). Thus, to predict the effects of participant variants on element accessibility, we used our trained model to generate cell-type specific SNP accessibility difference (SAD)[120] scores.

Our peak-centric approach successfully re-identified the HCFP1 cRE2 SNVs that we reported to be pathogenic for CFP[34], and scATAC data revealed that cRE2 was accessible in cMN7 at mouse e10.5 but not e11.5 (Fig. 7a). Examining cRE2 SNV SAD scores, we found that all four Cluster A LoF variants were predicted to close the chromatin (SAD Z-scores of −4.88, −3.60, −6.29, and −3.93). Moreover, these predicted variant effects were specific to cMN7 at e10.5 (but not e11.5, Fig. 7b), further underscoring the importance of accurately parsing both cell type and developmental cell state. We then experimentally corroborated the predicted variant effect on chromatin accessibility by performing scATAC on two CRISPR-mutagenized mouse lines harboring HCFP1 cRE2 Cluster A SNVs (*cRE2^{Fam5/Fam5}* and *cRE2^{Fam4/Fam4}* mouse models)[34]. Consistent with our machine-learning predictions, we observed subtle yet consistent reductions in *cis* chromatin accessibility for both mutant lines when compared to wildtype (4/4 replicates total; mean normalized mutant/wildtype coverage = 0.59; Fig. 7c). We also found positive evidence for site-specific footprinting overlapping the cRE2 NR2F1 binding site in wildtype, but not in the two mutant lines (Fig. 7b, d), consistent with results from targeted antibody-based assays[34]. Finally, to circumvent batch and normalization effects across

## Table 1 | Non-coding candidate variants and putative target genes

| Pedigree | CCDD | Sporadic vs familial | Isolated vs syndromic | Monoallelic vs biallelic variant(s) | Non-coding variant (hg38) | Peak Type | Nearest gene | Target gene | Distance to target (kb) | Reporter ID | Peak to gene r | Peak to gene FDR | gnomAD allele frequency | Predicted mechanism | SAD Z-score | Target gene loeuf[a] | Target gene pHaplo[b] | Target gene pTriplo[c] | Non-coding Z-score[d] |
|---|---|---|---|---|---|---|---|---|---|---|---|---|---|---|---|---|---|---|---|
| S25 | CFEOM | S | I | M | chr10:129794079 TTGAG>T | D | EBF3 | EBF3'(±DRS) | 170 | hs2776 | 0.24 | 2.90E-07 | 8.37E-05 | LoF | -11.77 | 0.15 | 1.00 | 1.00 | 3.10 |
| S176 | MGJW | F, AD | I | M | chr10:129884231 C>A | I | EBF3 | EBF3'(±DRS) | - | hs2775 | 0.29 | 3.89E-10 | 4.88E-05 | GoF | 0.11 | 0.15 | 1.00 | 1.00 | 3.74 |
| S95 | Ptosis | F, AD | Cr | M | chr10:129944464 G>C | I | EBF3 | EBF3'(±DRS) | - | hs2774 | 0.21 | 7.76E-06 | - | GoF | 0.98 | 0.15 | 1.00 | 1.00 | 5.14 |
| S12 | DRS | S | Ca | B(h) | chr11:72394626 C>G | I | CLPB | PHOX2A(+CFEOM) | 156 | - | 0.26 | 1.09E-08 | 1.41E-03 | GoF | 0.18 | 0.80 | 0.76 | 0.98 | 2.32 |
| S32 | Ptosis | F, AD | I | M | chr2:175005662 C>T[†] | P | CHN1 | CHN1(+DRS) | - | - | 0.48 | 1.31E-28 | 1.39E-04 | LoF | -0.38 | 0.57 | 0.41 | 0.72 | 2.59 |
| S251 | CFEOM/DRS | F, AD | I | M | chr2:175006051 GCTT>G[†] | P | CHN1 | CHN1(+DRS) | - | - | 0.48 | 1.31E-28 | - | GoF | 2.29 | 0.57 | 0.41 | 0.72 | 2.08 |
| S230 | DRS | S | I | M | chr20:40866929-40945626[††] | D | TOP1 | MAFB(+DRS) | 256 | hs2769 hs2770 | 0.23* | 1.19E-05* | - | - | - | 0.40 | 0.94 | 1.00 | 2.19* |
| S205 | CFP | S | I | B(ch) | chr5:51172762 T>A | D | ISL1 | ISL1 | 221 | hs1321 | 0.74 | 1.36E-86 | 2.26E-03 | LoF | -0.41 | 0.23 | 0.95 | 0.85 | -2.28 |
| S205 | CFP | S | I | B(ch) | chr5:51172961 T>G | D | ISL1 | ISL1 | 221 | hs1321 | 0.74 | 1.36E-86 | 2.33E-03 | LoF | -0.12 | 0.23 | 0.95 | 0.85 | -2.28 |
| S190 | DRS | S | I | B(ch) | chr22:27493955-27497536[†,††] | D | MN1 | MN1 | 307 | hs2757 | - | - | 1.38E-04 | - | - | 0.48 | 0.99 | 0.92 | 0.29* |
| S238 | DRS | S DRS, F, AD SNHL | SNHL | B(ch) | chr22:27493955-27497536[†,††] | D | MN1 | MN1 | 307 | hs2757 | - | - | 1.38E-04 | - | - | 0.48 | 0.99 | 0.92 | 0.29* |
| S191 | DRS | S | I | B(ch) | chr17:1455690 G>A[††] | I | CRK | CRK | - | - | - | - | - | GoF | 0.44 | 0.34 | 0.97 | 1.00 | 0.30 |
| S191 | DRS | S | I | B(ch) | chr17:1456361 G>A[††] | P | CRK | CRK | - | - | - | - | 1.51E-03 | LoF | -1.24 | 0.34 | 0.97 | 1.00 | - |
| S211 | DRS | S | I | B(ch) | chr17:1455565 C>T[††] | I | CRK | CRK | - | - | - | - | 1.19E-04 | GoF | 0.49 | 0.34 | 0.97 | 1.00 | 0.30 |
| S211 | DRS | S | I | B(ch) | chr17:1456436 G C>G[††] | P | CRK | CRK | - | - | - | - | 3.77E-04 | LoF | -12.28 | 0.34 | 0.97 | 1.00 | - |
| S211 | DRS | S | I | B(ch) | chr17:1456438 G>A[††] | P | CRK | CRK | - | - | - | - | 3.77E-04 | LoF | -2.06 | 0.34 | 0.97 | 1.00 | - |
| WL | DRS | F, 2 sibs | I | M | chr17:48003752 A>C[††] | D | CDK5RAP3 | CDK5RAP3 | 22 | hs2777 | 0.57 | 8.04E-43 | - | GoF | 4.31 | 0.97 | 0.24 | 0.54 | 1.94 |
| S174 | MBS | S | I | B(ch) | chr17:48003557 C>G[††] | D | CDK5RAP3 | CDK5RAP3 | 22 | hs2777 | 0.57 | 8.04E-43 | 4.04E-03 | LoF | -0.15 | 0.97 | 0.24 | 0.54 | 1.94 |
| S174 | MBS | S | I | B(ch) | chr17:48003826 C>T[††] | D | CDK5RAP3 | CDK5RAP3 | 22 | hs2777 | 0.57 | 8.04E-43 | 9.42E-04 | GoF | 1.69 | 0.97 | 0.24 | 0.54 | 1.94 |
| S156 | CFP | F, AD | I | M | chr3:128459417 G>C[††] | D | DNAJB8 | GATA2 | 7 | - | 0.28 | 6.08E-10 | - | LoF | -4.88 | 0.34 | 0.98 | 0.87 | - |
| S180 | CFP | F, AD | I | M | chr3:128459454 A>G[††] | D | DNAJB8 | GATA2 | 7 | - | 0.28 | 6.08E-10 | 3.95E-05 | GoF | 2.88 | 0.34 | 0.98 | 0.87 | - |
| S194 | CFP | F, AD | I | M | chr3:128459455 G>A[††] | D | DNAJB8 | GATA2 | 7 | - | 0.28 | 6.08E-10 | - | GoF | 11.40 | 0.34 | 0.98 | 0.87 | - |

S sporadic, F familial, AD autosomal dominant inheritance pattern, SNHL sensorineural hearing loss, M monoallelic, B biallelic, h homozygous, ch compound het, ± denotes that reported individuals with variants in the gene have a syndromic phenotype and a subset also have the stated CCDD phenotype, + denotes an established CCDD gene for stated phenotype, I isolated, P promoter, D distal, LoF loss-of-function, GoF gain-of-function, I intronic, Cr craniofacial, Ca cardiac

aCoding loss-of-function intolerance—https://doi.org/10.1038/s41586-020-2308-7
bCoding dosage sensitivity—https://doi.org/10.1016/j.cell.2022.06.036
cNon-coding mutational constraint (1 kb windows)—https://doi.org/10.1101/2022.03.20.485034
*mean value across deleted interval
†Multi-hit gene; ††non-coding deletion

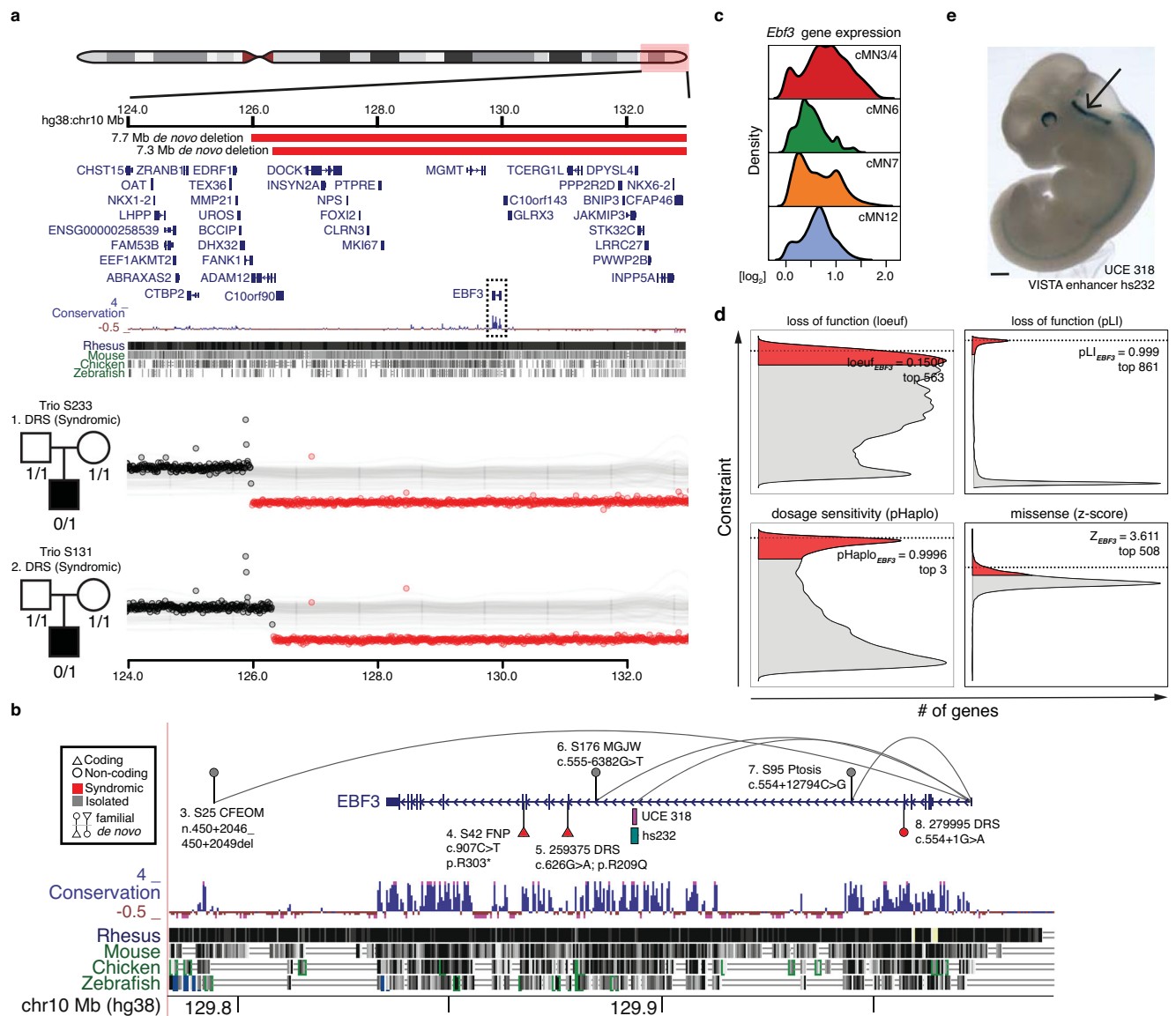

**Fig. 5 | An integrated coding/non-coding candidate allelic series for *EBF3*.**
**a** Window depicting the terminal arm of chr10q (top). Large de novo deletions in two trios (middle, bottom) with simplex syndromic DRS (S233 and S131) overlap multiple coding genes including *EBF3* (boxed), an exceptionally conserved gene at the coding and non-coding level. **b** Nominated coding and non-coding SNVs and indels connected to *EBF3*. For each variant, the subject's WGS ID code, CCDD phenotype, and the variant coordinate in NG_030038.1 (and if coding or non-coding and if familial or de novo) is indicated. Variants 5 and 8 are reported previously in DECIPHER and elsewhere (https://doi.org/10.1016/j.ajhg.2016.11.020). Peak-to-gene links containing variants connected to *EBF3* are depicted by curved lines. *EBF3* contains highly conserved non-coding intronic elements, including ultraconserved element UCE318 in intron 6, whose sequence drives strong expression in the embryonic hindbrain. **c** Imputed gene expression profiles for

*Ebf3*. **d** *EBF3* is exceptionally intolerant to loss-of-function, gene dosage, and missense variation. Density plots depict the genome-wide distribution of loss-of-function constraint (loeuf, pLI) (https://doi.org/10.1038/s41586-020-2308-7; https://doi.org/10.1038/nature19057), probability of haploinsufficiency (pHaplo) (https://doi.org/10.1016/j.cell.2022.06.036), and missense constraint (z-score) (https://doi.org/10.1038/ng.3050). Respective scores exceeding thresholds of 0.35, 0.9, 0.84, and 2.0 are colored red. *EBF3* (dotted lines) ranks as the 563rd, 861st, 3rd, and 508th most constrained gene in the genome, respectively. Distributions are rescaled for consistent signs and ease of visualization. **e** Lateral view of in vivo reporter assay testing UCE318 (VISTA enhancer hs232; *n* = 7 embryos), a putative *EBF3* enhancer (peak-to-gene *r* = 0.42, FDR = 6.72 × 10⁻²²). Strong reporter expression is observed in the embryonic hindbrain (arrow). Scale bar in **e** = 500 μm and is approximate based on E11.5 embryo average crown-rump length of 6 mm.

separate experiments, we performed scATAC on embryos from wildtype-by-mutant crosses from $cRE2^{Fam5/Fam5}$ and directly measured the resultant heterozygous mutant allele fraction in *cis* (binomial ATAC, Fig. 7e). This approach generates an internally calibrated estimate of effect size and is sufficiently powered to detect true differences at relatively low sequencing coverage (i.e., chromatin accessibility profiles of rare or transiently developing cell types). We found a significant depletion of *Fam5* mutant alleles across multiple replicates, again consistent with a LoF mode of pathogenicity

(wildtype/mutant counts = 4.2; mean probability of success = 0.81; 95% CI[0.73–0.87]; p value = 2.4 × 10⁻¹⁴; binomial test). These multiple lines of evidence, both at the epigenome-wide level and at a well-characterized individual locus, provide support that our machine-learning model is well-calibrated and not overfitted.

We next examined the predictions of the neural net at the epigenome-wide level, and among our 5353 cell type-aware candidate SNVs/indels, we identified 114 additional variants with normalized absolute SAD Z-scores >2; that is, variants predicted to significantly

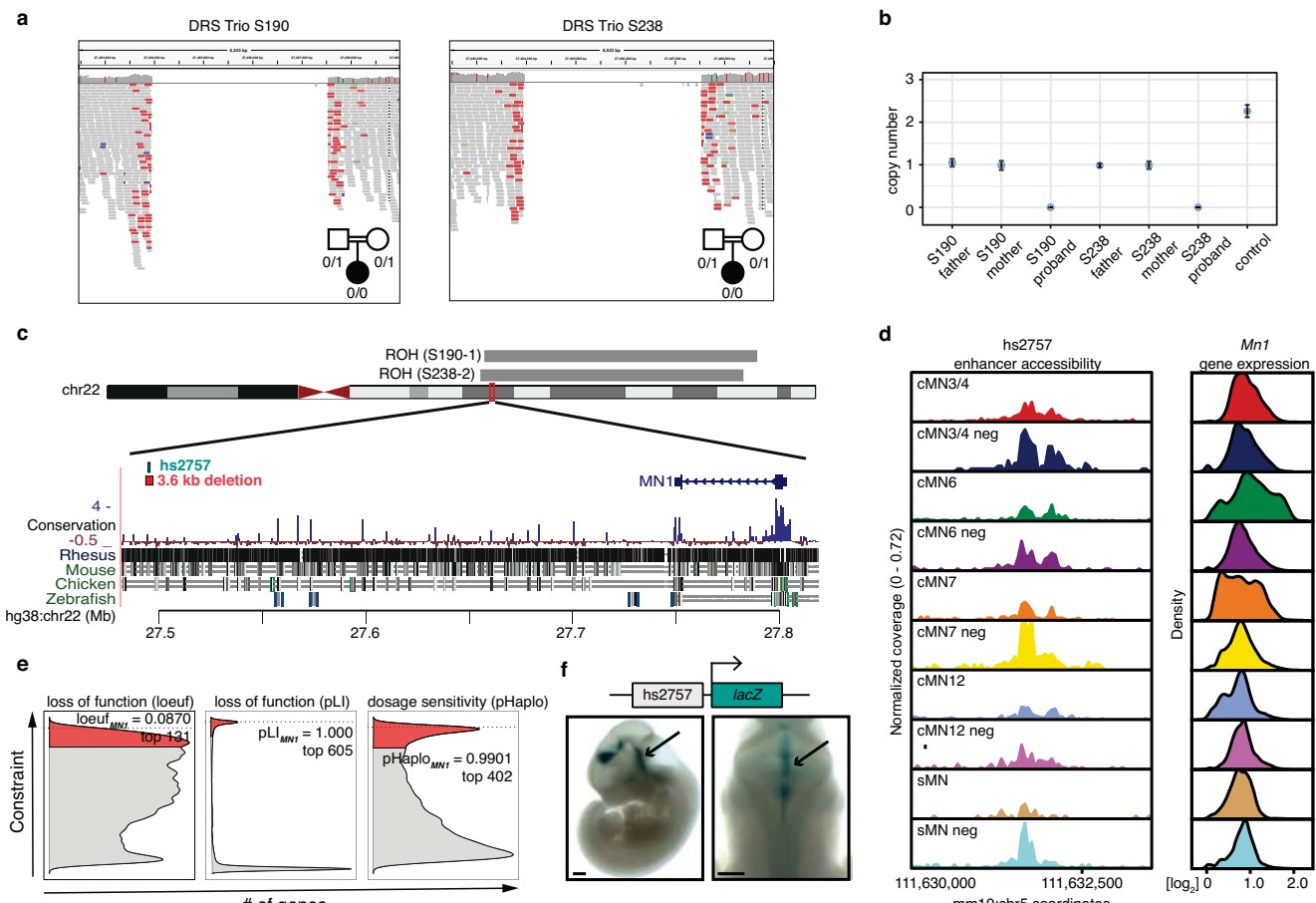

**Fig. 6 | _MN1_ enhancer deletions across multiple CCDD pedigrees. a** IGV screenshot depicting 3.6 kb non-coding deletions in two probands with DRS from separate consanguineous pedigrees (S190, S238). **b** ddPCR copy number estimates of deletions. For each pedigree, the affected proband is homozygous recessive for the deletion with one heterozygous allele inherited from each parent. Error bars denote 95% confidence intervals about the mean value estimated from two technical replicates per data point. **c** Genomic context of the non-coding deletions. The deletions (red bar below chr22 ideogram) fall within extended runs of homozygosity (gray bars above ideogram, 19.5 Mb, 18.8 Mb, respectively, of which 16 kb surrounding the deletion is shared between the probands) and eliminate putative enhancer hs2757 (green bar below ideogram) located 307 kb from nearest gene _MN1_. **d** hs2757 chromatin accessibility (left) and _Mn1_ imputed gene expression (right) profiles in the cMNs and surrounding cell types. _Mn1_ is widely expressed across multiple midbrain/hindbrain cell types, and hs2757 is accessible across multiple cell types, including cMN6. **e** Density plots depicting genome-wide

distribution of loss-of-function constraint (loeuf, pLI) (https://doi.org/10.1038/s41586-020-2308-7; https://doi.org/10.1038/nature19057), and probability of haploinsufficiency (pHaplo) (https://doi.org/10.1016/j.cell.2022.06.036) metrics. Respective scores exceeding thresholds of 0.35, 0.9, 0.84, and 2.0 are colored red. _MN1_ (dotted lines) ranks as the 131st, 605th, and 402nd most constrained gene in the genome, respectively. Distributions are rescaled for consistent signs and ease of visualization. **f** In vivo reporter assay testing hs2757 enhancer activity (humanized sequence, _n_ = 6 embryos). Lateral (left) and dorsal (right) whole mount _lacZ_ staining reveals that hs2757 consistently drives expression in midbrain and hindbrain tissue, including the anatomic territory of cMN6. Scale bar = 500 μm and are approximate measurements based on E11.5 embryo average crown-rump length of 6 mm. Scale bars in **f** = 500 μm and are approximate measurements based on E11.5 embryo average crown-rump length of 6 mm. Source data are provided as a Source Data file.

increase or decrease accessibility in _cis_ within their disease-relevant cellular context, including seven variants linked to multi-hit genes (Supplementary Data 9). When incorporating these SAD scores, we identified several cell type-aware candidate variants and peaks with convergent lines of evidence. Several of the non-coding variants connected to known CCDD genes had significant SAD scores (Table 1). The variant connected to _CHN1_ segregated in a parent and child with a mixed CFEOM-DRS phenotype was predicted to increase accessibility (SAD Z-score = +2.29). This is notable because _CHN1_ coding variants result in atypical DRS through a gain-of-function mechanism[23,52,121]. The _EBF3_ non-coding variants chr10:129794079 TTGAG > T, chr10:129884231 C > A, and chr10:129944464 G > C had SAD scores of −11.77, +0.11, and +0.98, respectively. In addition, compound heterozygous variants in two DRS probands in the multi-hit promoter region of _CRK_, a gene which has known roles in sMN axon growth and neuromuscular junction formation but no known link to human Mendelian

phenotypes, had significant negative scores consistent with LoF (SAD Z-scores = −13.69, −2.06; Supplementary Data 8). Such highly annotated non-coding variants are attractive candidates for downstream functional validation, as they deliver distinct, refutable predictions for gene targets, cell types, and effects on accessibility.

**Nominated cell type-specific variants alter expression in vivo**
Although we show that single-cell chromatin accessibility is a strong predictor of cMN enhancer activity, even highly conserved and presumably functional enhancers can be surprisingly robust to mutagenesis[8,122–124]. Therefore, to evaluate the functional consequences of our nominated CCDD variants, we selected 33 elements harboring cell type-aware candidate SNVs for in vivo humanized enhancer assays. For testing, we prioritized these variants based on multiple annotations from our framework, including conservation, significant SAD scores, multi-hit peaks/genes, and cognate gene

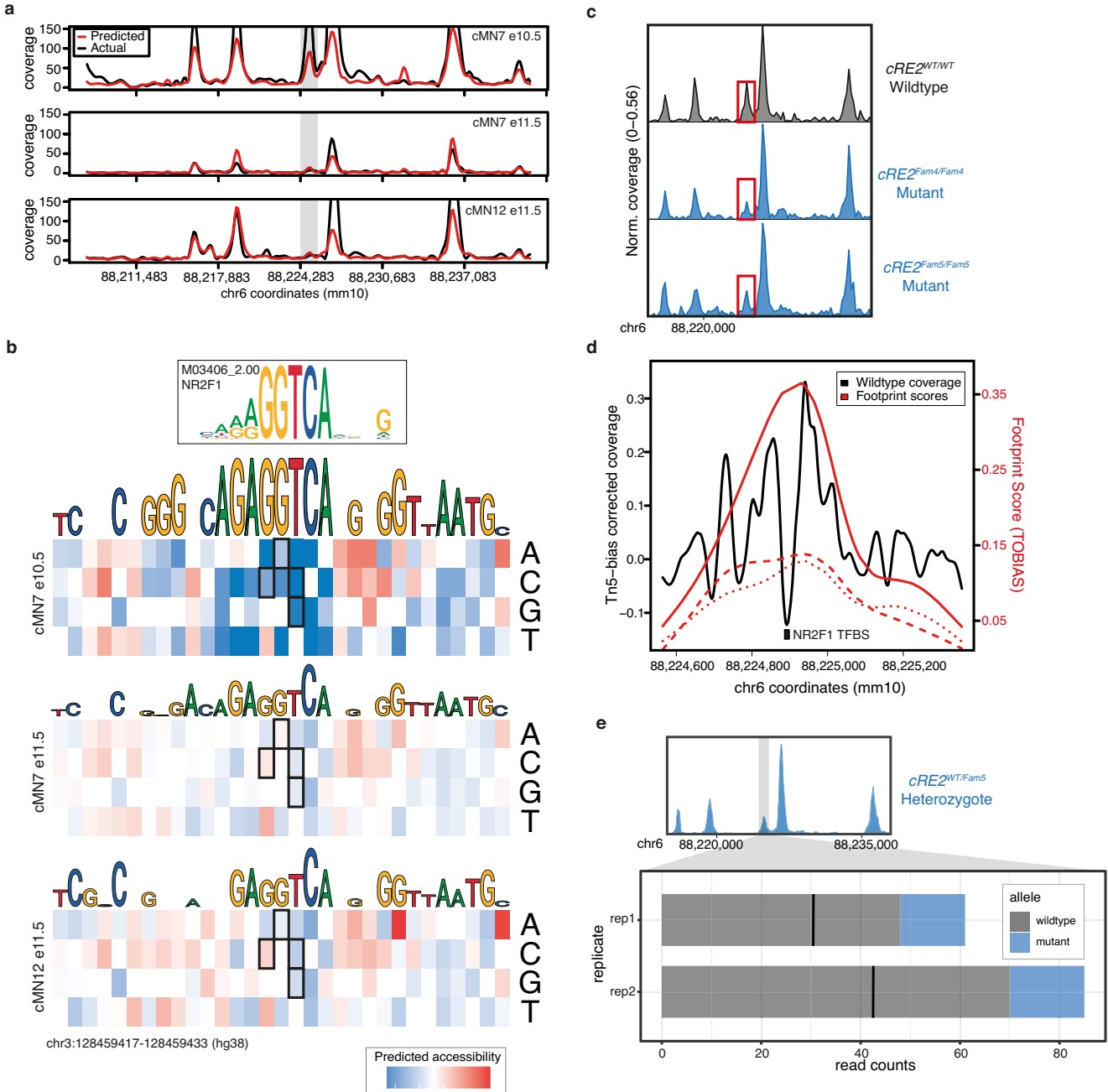

**Fig. 7 | scATAC-trained convolutional neural network accurately predicts cell type-specific accessibility status and human mutation effects in a transiently developing cell type. a** Neural net predicted chromatin accessibility profiles (red) compared to actual scATAC sequencing coverage (black) for a region of mouse chromosome 6 in three cell types (cMN7 e10.5, cMN7 e11.5, and cMN12 e11.5). The gray box highlights a transient 678 bp peak (cRE2) that is accessible in cMN7 e10.5, but not cMN7 e11.5 or cMN12 e11.5. SNVs within the human orthologous peak cRE2 cause congenital facial weakness, a disorder of cMN7. **b** Neural net-trained in silico saturation mutagenesis predictions for specific nucleotide changes in human cRE2 for cMN7 e10.5, cMN7 e11.5, and cMN12 e11.5. Predicted loss-of-function nucleotide changes are colored in blue and gain-of-function in red. Predictions for four known loss-of-function pathogenic variants (chr3:128178260 G > C, chr3:128178261 G > A, chr3:128178262 T > C, chr3:128178262 T > G) are boxed. All four pathogenic variants are predicted loss-of-function for cMN7 e10.5, but not cMN7 e11.5 or cMN12 e11.5. **c** Pseudobulk accessibility profiles of *cRE2* (red box) CN7 e10.5 for wildtype and two CRISPR-mutagenized mouse lines (*cRE2^Fam4/Fam4^ and cRE2^Fam5/Fam5^*) show a

qualitative reduction in cRE2 scATAC sequencing coverage, consistent with in silico saturation mutagenesis predictions. Each pseudobulk profile represents normalized sequencing coverage across two biological replicates. **d** Locus-specific footprinting evidence overlapping cRE2. A 792 bp window showing sequencing coverage for cMN7 e10.5 after correcting for Tn5 insertion bias. The NR2F1 transcription factor binding site is mutated in individuals with HCFP1-CFP and overlaps a local minimum in scATAC coverage. TOBIAS footprinting scores for *cRE2* wildtype, *cRE2^Fam4/Fam4^*, and *cRE2^Fam5/Fam5^* are depicted in solid, dashed, and dotted lines, respectively. **e.** Stacked barplot depicting wildtype (gray) versus mutant (blue) scATAC read counts over a 7.7 kb window for cMN7 e10.5 in *cRE2^WT/Fam5^* heterozygote embryos. cRE2 mutant alleles are consistently depleted across two biological replicates (counts_WT / counts_MUTANT = 4.21; $p$ value = $2.4 \times 10^{-14}$, binomial test). The mean expected value under the null is depicted by a solid black line for each experiment. Scale bars in **f** and **i** = 500 μm and are approximate measurements based on E11.5 embryo average crown-rump length of 6 mm. Source data are provided as a Source Data file.

predictions (Supplementary Fig. 10 and Supplementary Data 10). We first screened the wildtype human enhancer sequences and detected positive enhancer activity in 82% (27/33) of candidates (210 embryos total, Methods). Combining these with the 26 previously tested, we found enhancer activity in 44/59 total (75%; 366 embryos total). Importantly, we note that these elements were not selected randomly and, therefore, not intended to reflect generalizable patterns across the genome.

Next, we tested 4 of the 27 positive elements by introducing the nominated CCDD SNVs into the wildtype sequence. Remarkably, one mutant enhancer harboring multiple candidate variants for DRS and MBS (hs2777-mut; 15 embryos total) showed a visible gain of expression compared to wildtype (hs2777; 11 embryos total), including in midbrain, hindbrain, and neural tube (Supplementary Fig. 11a, b). Wild-type hs2777 is accessible across multiple cell types and has peak-to-gene links to seven genes (*Cdk5rap3, Nfe2l1, Sp2, Tbx21, Npepps, Socs7,* and *Snx11*), and ABC enhancer prediction for *Cdk5rap3,* specifically to cMN7 at e10.5. hs2777-mut contains four SNVs (1 DRS, 2 MBS, 1 off-target, mutating 0.21% of original wildtype base pairs; Supplementary Fig. 11c, d). To better decompose the individual effects of these variants, we performed in silico saturation mutagenesis across the entire hs2777 sequence (Supplementary Fig. 11e). We observed notable gain-of-function effects for two of the three on-target SNVs (DRS Variant C, and MBS Variant D; hg38 chr17:48003826 C > T and chr17:48003752 A > C, respectively) within the affected cell types, with corresponding SAD Z-scores ranging from +1.12 to +4.34.

## Discussion

We have developed a publicly available atlas of developing cranial motor neuron chromatin accessibility and have combined it with cell type-specific histone modification and in vivo transgenesis information to generate a reference set of enhancers with cognate gene predictions in a set of rare, transiently developing cell types. Such a resource can be used to discover highly specific cREs and target genes underlying the molecular regulatory logic of cMN development. Furthermore, we can leverage known properties of the cMNs to inform comparative studies across diverse cell types. For example, the ocular cMNs are known to be selectively resistant to degeneration (compared to sMNs) in diseases such as ALS. Therefore, understanding the differentially accessible cREs that underlie differences between cMNs/sMNs could render important clues to the mechanisms of selective resistance/vulnerability and ultimately open potential therapeutic avenues[90]. Finally, a deeply sampled, highly specific chromatin accessibility atlas may help to learn generalizable features that predict enhancer activity in additional cell types. Importantly, cranial nerve expression is a core readout for tested cREs in the VISTA enhancer database, thereby providing invaluable ground truth data at an overlapping developmental timepoint (e11.5)[66].

We used this reference to nominate and prioritize non-coding variants in the CCDDs, a set of Mendelian disorders altering cMN development, and demonstrate that principled prioritization approaches can select appropriate candidates for downstream functional validation (e.g., transgenic reporter assays, non-coding in vivo disease models, etc.), which are otherwise often costly and labor-intensive with high rates of failure. To aid in interpretation, we connected non-coding variants to their cognate genes using imputed gene expression values from separate assays (diagonal integration). This approach allowed us to leverage existing information on cognate coding genes, including known disease associations and coding constraints[92]. Moreover, such integrated cell type-aware datasets provide important context to cell type-agnostic estimates of non-coding constraints (discussed in ref. [125]). When applying this framework to our CCDD cohort, we achieved a search space reduction of four orders of magnitude (i.e., $5.5 \times 10^7$ reduced to $5.4 \times 10^3$ searchable variants) making non-coding candidate sets human-readable and tractable for functional and

mechanistic studies (23.6 candidates per monoallelic pedigree; 13.6 per biallelic pedigree). Furthermore, we incorporated multiple lines of evidence such as allelic aggregation, cognate gene identification, mutational constraint, and functional prediction. This approach successfully re-identified the pathogenic variants in our cohort at the *GATA2* cRE2 locus[34] and led us to nominate additional disease variants (Table 1). We also identified compelling individual candidate variants and peaks without multiple hits. Such candidates will be easier to resolve with larger cohort sizes and larger families. Indeed, our ability to reduce candidate variant numbers was limited by the large proportion of unsolved small dominant pedigrees in our cohort, which are notoriously difficult to analyze. Moreover, while de novo and recessive mutations are clearly an important source of causal pathogenic variation in sporadic cases, such cases are also more likely to involve non-genetic etiologies.

Although a given peak can harbor hundreds of predicted transcription factor binding motifs, we demonstrate in principle that locus-specific footprinting can implicitly reduce a ~1 kb peak to a ~10 bp individual transcription factor binding site of interest. Given sufficient sequencing coverage[126] and data quality, such approaches could immediately be applied to other rare diseases and cell types. Alternatively for common diseases where causal non-coding variants are more abundant, locus-specific footprinting (in concert with careful demarcation of element boundaries, chromatin accessibility QTL analysis[127], and statistical fine-mapping[128]) may further resolve causal common variants and identify affected transcription factor binding sites across the genome—all inferred from a single assay. Proof of feasibility of such approaches in rare diseases could also influence data collection strategies for common diseases[129].

Through our analysis, we also encountered potential limitations affecting non-coding variant interpretation. We, in part, leveraged sequence conservation and constraint to prioritize pathogenic variants. However, while the known genes and cREs underlying cMN development are highly conserved, a conservation-based strategy may not identify pathogenic variants in human-specific and/or rapidly evolving sequences[124,130,131]. Strikingly, we also found that even relatively subtle differences in cellular composition and ATAC/RNA collection strategies can distort cognate gene estimates. These findings should inform appropriate sampling strategies in the future, such as single-cell multiomic assays. Unbiased genetic strategies such as partitioned LD score regression can be useful for defining disease-relevant cell types, though such approaches are effectively restricted to common diseases[132]. Moreover, we find that even when sampling the appropriate cell type, subtle differences in cell state can profoundly influence variant interpretation. We provide a concrete example at the well-characterized non-coding *GATA2* locus[34], where pathogenic variant effects are no longer detectable in the same cell type within a mere 24 hours of development (i.e., embryonic day 10.5 versus 11.5). Moreover, we sampled cMNs at e10.5 and e11.5 based on developmental patterns of previously described protein-coding mutations, but we do not exclude the possibility that disease mutations may also be relevant at different timepoints. Therefore, while our genetic framework can generalize to other disorders, we suspect that appropriate prospective or retrospective epigenomic cell sampling will benefit from highly detailed biological knowledge of each specific disease process.

Finally, the interpretation of non-coding variants can benefit from our knowledge of coding variants as they share challenges in common —namely, practical limitations in allelic expansion and functional validation. Here, we present generalizable approaches that aggregate plausible alleles based on physical (peak-centric) and biological (gene-centric) proximity to facilitate allelic expansion in a principled manner. These challenges may be further alleviated by expanding rare disease data-sharing platforms[133] to more comprehensively incorporate non-coding variation. Finally, the development of functional perturbation assays that balance both scalability[123] and specificity[134] will

disproportionately benefit the validation of non-coding variants, which are naturally more abundant and cell type-specific than coding variants. The outputs of such assays would also iteratively provide training material for further refined functional prediction algorithms.

Rapid advances in next-generation sequencing technologies have led to a renaissance in Mendelian gene discovery. As access to WGS and functional genomics data becomes less limiting, alternative analytical and experimental frameworks will be needed to finally resolve Mendelian cases and disorders that are otherwise recalcitrant to traditional exome-based approaches.

## Methods

Research participants were enrolled in the long-term genetic study of CCDDs at Boston Children's Hospital (BCH; clinicaltrials.gov identifier NCT03059420). The Institutional Review Board at BCH approved the study (protocol 05-03036R, 02-05-070R, 99-05-085). Informed consent was obtained from each participant or legal guardian. Individual-level data were de-identified, and studies were performed in compliance with US 45.CFR.46 and the Declaration of Helsinki.

### Mouse husbandry, dissection, dissociation, FACS

Mice were maintained in pathogen-free environments and fed ad libitum with a sterile standard diet and water in temperature, humidity, and light-controlled rooms (22 °C set-point ±l.3 °C, RH35-70% ±5%, 12/ 12 light/dark cycle, 10–15 air changes per hour). We performed husbandry, dissection, dissociation, and FACS[135] in accordance with the Institutional Animal Care and Use Committees of Boston Children's Hospital with IRB approval (BCH Institutional Care and Use Committee Protocol #00001852). Embryo sex was not determined. We crossed C57BL/6J (JAX # 000664) female mice with either 129S1/C57BL/6J *Isl1$^{MN}$*:GFP (JAX # 017952[35]) or *Hb9*:GFP (JAX # 005029[36]) male mice and separated them following one night of breeding. Pregnant females were sacrificed at 10.5 or 11.5 days post-conception, and whole embryos were grossly dissected in chilled 1x PBS (Thermo Fisher) and then immediately placed in 1x B27 supplement (Gibco 17504044) in Hibernate E (Fisher NC0285514). Next, GFP-positive cranial motor neurons, GFP-positive spinal motor neurons, and GFP-negative surrounding cells were microdissected in pre-chilled HBSS (Thermo Fisher) and placed in 1x B27 supplement, 1x Glutamax (Thermo Fisher 35050061), and 100 U/mL Penicillin-Streptomycin (PenStrep, Thermo Fisher 15140122) in Hibernate E (medium 2). Microdissected tissues were dissociated using papain and ovomucoid solutions prepared from the Papain Dissociation System (Worthington Biochemical LK003150). Tissues were resuspended in papain solution. Samples were then incubated at 37 °C for 30 min and agitated every 10 min to ensure complete dissociation. Following incubation, samples were spun down at 300×*g* for 5 min, the supernatant was removed, and dissociated tissues were resuspended in 500 uL of ovomucoid solution (plus or minus 100 μL depending on the quantity of tissue). Tissues were again spun down at 300×*g* for 5 min and resuspended in 500 μL of medium 2 (plus or minus 100 μL depending on the quantity of tissue) and transferred to a 5 mL polystyrene round bottom tube on ice. Live GFP-positive singlets were separated from GFP-negative cells (GFP-negative limb buds from embryos used as a negative control to set FACS gates) using an ARIA-561 FACS machine at the Immunology Research Core at Harvard Medical School (for ATAC-seq samples), and a BD FACS Aria II at the Jimmy Fund Core at the Dana-Farber Cancer Institute (for bulk and single-cell RNA-seq samples). GFP-positive cells were collected either into 200 uL of media containing 1x Glutamax, 100 U/mL PenStrep, and 2% 2-Mercaptoethanol (Gibco 21985023) in Neurobasal-A Medium (Thermo Fisher 10888022) for ATAC-seq, or into 96 well fully-skirted Eppendorf plates containing a starting volume of 5 ul/well of Hibernate E for single cell RNAseq, or directly into 1.5 ml tubes containing Qiagen RNeasy Lysis buffer/Buffer RLT (Qiagen 79216) for the bulk RNAseq. Embryos were not selected based on sex. Embryos were excluded if they did not match the expected developmental stage as estimated from morphological features. All biological replicates represent distinct samples except when noted ("technical replicate").

### Single-cell ATAC-seq: nuclei isolation, tagmentation, and sequencing

We performed fluorescence-assisted microdissection to collect samples cMN3/4, cMN7, and sMN from *Isl1$^{MN}$*:GFP mice and likewise to collect samples of cMN6, cMN12, and sMN from *Hb9*:GFP mice, each at both e10.5 and e11.5. We performed FACS-purification as described above to collect GFP-positive motor neurons, as well as GFP-negative cells surrounding the motor neurons, to better distinguish between motor neuron versus non-motor neuron regulatory elements (for a total of 20 sample types, nine with biological replicates and two with technical replicates for 32 samples in all). Nuclei were isolated in accordance with Low Cell Input Nuclei Isolation guidelines provided by Demonstrated Protocol–Nuclei Isolation for Single Cell ATAC Sequencing Rev A (Protocol #CG000169) from 10x Genomics. Cell suspensions were spun down at 300×*g* for 5 min at 4 °C in a fixed angle centrifuge, the supernatant was removed, and the pellet was resuspended in 50 uL of 0.04% BSA in PBS. The cell solution was then transferred to a 0.2 mL tube and centrifuged at 300×*g* for 5 min at 4 °C in a swinging bucket centrifuge. Without contacting the bottom of the tube, 45 uL of supernatant was removed, and the cell pellet was resuspended in 45 uL of chilled Lysis buffer (10 mM Tris-HCl (pH 7.4), 10 mM NaCl, 3 mM MgCl$_2$, 0.1% Tween-20, 0.1% Nonidet P40 Substitute, 0.01% Digitonin, 1% BSA, in nuclease-free water). Nuclei suspensions were incubated on ice for 3 min and 50 uL of wash buffer (10 mM Tris-HCl (pH 7.4), 10 mM NaCl, 3 mM MgCl$_2$, 1% BSA, 0.1% Tween-20, in nuclease-free water) was added to the suspensions without mixing. Nuclei suspensions were then spun down in a swinging bucket centrifuge at 500×*g* for 5 min at 4 °C, 95 uL of supernatant was removed, and 45 uL of nuclei buffer was added. Samples were again spun down in a swinging bucket centrifuge at 500×*g* for 5 min at 4 °C, all supernatant was removed without contacting the bottom of the tube, and nuclei were resuspended in 7 uL of nuclei buffer. About 2 uL of this final nuclei suspension was added to 3 uL of nuclease-free water, and 5 uL of trypan blue, and cell viability was inspected using the Countess II FL Automated Cell Counter (Thermo Fisher Scientific AMQAF1000). We performed scATAC transposition, droplet formation, and library construction from protocol CG000168 using v1 reagents (10x Genomics). scATAC libraries were sequenced on the Illumina NextSeq 500 system using standard Illumina chemistry. Paired inserts were a minimum of 2 × 34 bp in length excluding indices, and libraries were distributed to achieve an estimated coverage of ≥25,000 read pairs per cell in accordance with 10x Genomics guidelines (actual mean coverage was 48,772 reads per cell). Samples failing quality control were excluded (e.g., failed TapeStation output).

### scATAC preprocessing, peak calling, dimensionality reduction, and cluster analysis

**Preprocessing.** We performed a modified workflow based on ref. 59. Briefly, we generated fastq files from bcl using cellranger *mkfastq*. We initially included all single-cell ATAC barcodes perfectly matching an allowlist provided by 10x Genomics. We also included fixed barcodes if they had a maximum Hamming distance of 1 and if they were present in the top 2% of barcode counts. As a final check, we manually inspected the distribution of fixed barcodes in reduced dimension space to ensure a roughly even distribution across all cells. We aligned individual samples to the mm10 reference genome using Bowtie2[136], generated sample level.bam files, filtered reads with MAPQ <10, and performed PCR deduplication. We established heuristic coverage per cell thresholds for each sample separately. To generate cell counts, we

performed hard filtering based on log10[nfrags/barcode] for each sample separately.

**LSI clustering.** We performed LSI-based clustering to generate sample-level clades[59]. In order to enrich peak representation from rare neuronal populations, we manually assigned between 3 and 7 clades to each sample and then performed peak calling on each clade using MACS2[137]. We first performed cell QC based on heuristic filters (low FRiP and accessible peaks-per-cell outliers), then peak QC (filtering peaks in a low proportion of remaining cells per clade). All post-QC cells and peaks were then combined to generate a master peak-by-cell callset. Samples failing any stage of QC were excluded (e.g., inadequate read coverage).

**Dimensionality reduction.** We performed LSI-based dimensionality reduction (log-scaled TF-IDF transformation followed by singular value decomposition) on our binarized peak-by-cell matrix[59]. We used umap() (https://github.com/lmcinnes/umap) to further reduce the dimensionality of our data to three-dimensional UMAP coordinates. We then performed cluster analysis using Seurat's SNN-graph approach. Once the major clusters were defined, we repeated our dimensionality reduction and cluster analysis on each major cluster to generate subclusters. Finally, we calculated peak specificity scores[59] across all 23 major clusters, identifying 45,813 peaks.

**Cluster homogeneity, completeness, and purity**

In order to formalize the agreement between our dissection/FACS labels (class) and our cluster/subcluster labels (cluster), we calculated homogeneity $h$, completeness $c$, and Vmeasure $V_\beta$, using the *sabre* package[62]:

$$h = \begin{cases} 1 & \text{if } H(C|K) = 0 \\ 1 - \frac{H(C|K)}{H(C)} & else \end{cases} \tag{1}$$

$$H(C|K) = -\sum_{k=1}^{|K|}\sum_{c=1}^{|C|} \frac{a_{ck}}{N} \log\left(\frac{a_{ck}}{\sum_{c=1}^{|C|} a_{ck}}\right) \tag{2}$$

$$H(C) = -\sum_{c=1}^{|C|} \frac{\sum_{k=1}^{|K|} a_{ck}}{N} \log\left(\frac{\sum_{k=1}^{|K|} a_{ck}}{N}\right) \tag{3}$$

$$c = \begin{cases} 1 & \text{if } H(K|C) = 0 \\ 1 - \frac{H(K|C)}{H(K)} & else \end{cases} \tag{4}$$

$$H(K|C) = -\sum_{c=1}^{|C|}\sum_{k=1}^{|K|} \frac{a_{ck}}{N} \log\left(\frac{a_{ck}}{\sum_{k=1}^{|K|} a_{ck}}\right) \tag{5}$$

$$H(K) = -\sum_{k=1}^{|K|} \frac{\sum_{c=1}^{|C|} a_{ck}}{N} \log\left(\frac{\sum_{c=1}^{|C|} a_{ck}}{N}\right) \tag{6}$$

$$V_\beta = \frac{(1+\beta)hc}{(\beta h) + c} \tag{7}$$

Where $C$ is the set of dissection/FACS class labels; $K$ is the set of clusters or subclusters; $a_{ck}$ is the number of single cells belonging to class $c$ and cluster or subcluster $k$; $N$ is the total number of single cells; and $\beta$ is the ratio of weights attributed to $c$ and $h$ ($V_\beta$ is the weighted harmonic mean of $c$ and $h$). As $\beta$ becomes very large or very small, $V_\beta$ approaches $c$ and $h$, respectively. Here we set $\beta$ to 1. To compare across different biological categories (e.g., GFP-positive, GFP-negative, etc.), we generated homogeneity/completeness measurements across different

subsets of cells (N) and different definitions of $k$ (e.g., cluster, sub-cluster) and $c$ (e.g., sample and time), summarized in Supplementary Table 2.

To quantify the maximum cellular representation of each cluster/subcluster, we also generated a per-cluster purity metric, $p$:

$$p_k = \frac{\max(a_{ck})}{\sum_{k=0}^{K} a_k} \tag{8}$$

Finally, because biological replicates were taken from different batches, we used the distribution of cluster/subcluster membership of cells from each replicate as an instrument to test for batch effects. We assigned every cell within a cluster or subcluster to its experiment of origin and calculated pairwise correlation estimates between all combinations of biological replicates. As expected, cluster/subcluster membership was most strongly correlated among biological replicates and was not driven by individual replicates.

**Motif enrichment and aggregated footprinting analysis**

We used the mouse motifs from the cis-BP database from the chrom-VARmotifs database to compute cluster and sample-specific motif footprinting and enrichments (mouse_pwms_v2). For each motif, we identified all sites in peaks where a motif was present. Clusters 3, 4, 5, and 9 were excluded from footprint analysis. We next identified differentially accessible peaks for each group of interest using ArchR's getMarkerFeatures() function, normalizing for differences across groups with transcriptional start site (TSS) Enrichment and log10(n-Frags). We selected peaks for each group that met an FDR threshold of below 0.01 and a LogF2C of ≥1. Aggregated footprint plots were generated for select motifs using plotFootprints(), by first normalizing the Tn5-bias by subtracting it from the footprinting signal. For site-specific footprints, we used TOBIAS to generate Tn5-bias-corrected bigwigs and footprint scores across the genome for each cell type[138]. For bias estimation and correction we excluded ENCODE denylist regions from mm10-blacklist.v2.bed (https://github.com/Boyle-Lab/).

**In vivo *lacZ* enhancer validation**

We selected 26 putative wild-type enhancers for downstream experimental validation based on the following criteria. First, we selected elements with significant cell type specificity scores[59]. Next, we excluded any elements that did not lift over to the human genome (hg19). We then identified elements with evidence of H3K27Ac marks in any cell type from the ENCODE portal[67] and no existing experimental data in the VISTA enhancer browser[66] (freeze September 2019). Finally, we performed manual curation in order to select elements with high conservation, against elements in repetitive regions, and ensured the representation of elements from cMNs 3, 4, 6, 7, 12, and sMNs.

We performed in vivo enhancer testing using enSERT transgenesis[139]. Briefly, the orthologous human sequence of each candidate enhancer was cloned into a pCR4-Shh::lacZ-H11 vector (Addgene plasmid # 139098) containing the mouse *Shh* minimal promoter, *lacZ* reporter gene, and H11 safe harbor locus homology arms. The cloned construct, Cas9 protein, and H11-sgRNAs (Alt-R CRISPR-Cas9 tracrRNA, IDT, 1072532 and Alt-R CRISPR-Cas9 locus targeting crRNA, gctgatggaacaggtaacaa) were delivered via mouse embryonic pronuclear injection (mouse FVB/NJ JAX #001800) and transferred to female hosts. Embryos were collected at e11.5, stained with X-gal, and evaluated for reporter activity.

For candidate variant testing in 33 enhancers, we generated cloned elements bearing the human reference or variant allele as described above. In the case of compound heterozygous variants, we cloned both variants into the same construct in *cis*. In the case of full enhancer deletion candidates, we cloned only the wild-type enhancer.

## Bulk ATAC-seq

We performed bulk ATAC-seq[140] for FACS-purified cells from six anatomic/temporal regions: *Isl*[MN]:GFP-positive cMN3 at e10.5 and e11.5, cMN7 at e10.5, sMN e10.5 and e11.5, and *Isl*[MN]:GFP-negative hindbrain at e11.5. We processed the bulk ATAC sequencing data by running the fastq files through the Encode ATAC-seq pipeline (https://github.com/ENCODE-DCC/atac-seq-pipeline) using default parameters. To analyze peaks for each bulk sample, we used irreproducible discovery rate (IDR) optimal peaks, generated between pseudoreplicates or biological replicates when appropriate. After generating peaksets for each bulk sample, we created a bulk master peakset by concatenating all the individual peaksets and merging with bedtools *merge*. We further generated bulk peaksets specific to each sample using bedtools *subtract*, allowing for ≤50% overlap between peaks.

## Single-cell RNA-seq

The husbandry and collection strategy was identical to the scATAC strategy described above, except that we combined GFP-positive and -negative cells from the same dissections. We performed single-cell RNA-seq for FACS-purified eGFP-positive motor neurons from 6 anatomic/temporal regions: cMN3 + 4 and cMN7 from *Isl1*[MN]:GFP mice and cMN6 from *Hb9*:GFP mice, all at both e10.5 and e11.5 (for total of ten samples). In most samples, we spiked in 10% surrounding eGFP-negative hindbrain cells as an internal control for comparison to nearby cells that are not motor neurons. Samples were submitted to the Klarman Cell Observatory/Regev Lab at the Broad Institute of MIT and Harvard for processing on a 10X Genomics Chromium platform. The 10X Genomics Chromium Single Cell 3' Reagent Kit (using v2 single index chemistry, CG00052) was used for mRNA capture and library preparation. Samples were multiplexed for a read-depth goal of 50,000 reads/cell (actual mean coverage was 94,829 reads/cell). Sequencing was performed on a HiSeq 4000 by Broad Genomic Services using standard Illumina chemistry. The data were then aligned in the Engle lab using Cell Ranger v2.1.1 against the ENSEMBL *Mus musculus* genomic reference build GRCm38.87 (modified to include eGFP and tdTomato sequences). Quality control was performed in Seurat to remove doublets and low-read cells. Analysis was done in Seurat where samples were integrated with canonical correlation analysis (CCA)[141]. Motor neurons were identified from the expression of *eGFP, Isl1*, and other motor neuron markers (eGFP was regressed out to avoid affecting clusters).

## Bulk RNA-seq

We performed bulk RNA-seq for FACS-purified eGFP+ cells from seven anatomic/temporal regions: cMN3, cMN4, cMN6, cMN7 at each corresponding brainstem level, at both e10.5 and e11.5 (except for cMN6 that was only collected at e11.5 due to cell number limitations at e10.5; with two biological replicates from all times/regions and one additional technical replicate of cMN6, for a total of 15 samples). Samples from multiple litters were merged to reach a threshold for the appropriate cell number and sent to Rutgers RUCDR for library preparation and sequencing. For the e11.5 samples, 200 ng/sample of RNA was isolated with Oligo-dT beads, enriching for mRNA. Depletion of beta-globin mRNA and ribosomal RNA was performed. For the e10.5 samples and the e11.5 cMN6 samples, due to the lower total RNA from fewer starting cells in these nuclei at these ages, whole-transcriptome Nugen Amplification was performed. Samples were sequenced with a 100 bp paired-end strategy to sequence full-length transcripts on an Illumina HiSeq2500 for an approximate read-depth of 60 million paired-end reads/sample. This generated R1 and R2 reads for each of the two lanes of data/sample that were subsequently concatenated. STAR (Spliced Transcripts Alignment to a Reference)[142], a splice-aware tool, was used to align reads to ENSEMBL *Mus musculus* genomic reference build GRCm38.87, and RSEM (RNA-Seq by Expectation Maximization)[143] was used to generate the count files. We then used DESeq2[144] to make comparisons.

## Generating peak-to-gene links

**RNA integration.** For our original RNA inputs for peak-to-gene links, we performed scRNA-seq on cMN3 + 4, cMN6, and cMN7 dissections (GFP-positive and -negative) at e10.5 and e11.5. Our husbandry and collection strategy was identical to the scATAC strategy described above, except that we combined GFP-positive and -negative cells from the same dissections. We performed scRNA-seq protocol CG000168 using v2 single index chemistry and sequenced on the Illumina HiSeq 4000. To benchmark our scRNA-seq results, we also performed bulk RNAseq on cMN3, cMN6, and cMN7.

We integrated multiple scRNA-seq datasets from GFP-positive and -negative cells from cMN3/4, 6, and 7 dissections at e10.5 and e11.5 into a single Seurat object using Seurat's integration framework[141]. We excluded cells with more than 5% of reads aligning with the mitochondrial genome. After examining the distribution of the number of unique features and the number of unique reads per cell for each sample, we manually filtered cells with low feature counts. Finally, we normalized each sample using the NormalizeData() function, identified the top 10,000 variable features per sample, and scaled each sample using the ScaleData() function.

Next, we excluded scATAC clusters (clusters 3, 4, 5, and 9) with high proportions of GFP-positive sMN and cMN12 dissected cells, as those samples are not represented in our scRNA dataset. We then performed unconstrained scATAC-RNA integration on all remaining cells using addGeneIntegrationMatrix() in ArchR[80].

**Benchmarking imputed gene expression.** We then evaluated the projected gene expression values from our scATAC-RNA integration for three high-confidence scATAC clusters (cMN3/4.10, cMN6.6, and cMN7.2). We selected these clusters due to unambiguous sample membership based on microdissection origin (purity), FACS labels (corresponding to cMN7, cMN6, and cMN3/4, respectively), and known marker locus accessibility/expression. We compared imputed gene expression from these clusters to corresponding bulk RNAseq samples that were independently dissected and FACS-purified. Specifically, we performed differential expression analysis on bulk RNAseq data (DEseq v1.34.0[144]) and on imputed gene expression on scATAC-seq data (using getMarkerFeatures() function in ArchR). We fit a linear model of the $\log_2$[fold-change] expression for all combinations of bulk samples and single-cell clusters, and confirmed a significant positive correlation between projected gene expression for marker genes in each cluster against its corresponding bulk counterpart.

**Peak-to-gene parameters and benchmarking.** We calculated peak-to-gene correlations using ArchR's addPeak2GeneLinks() function, with reducedDims = IterativeLSI_ArchR. We included all high-confidence links (FDR <0.0001) with a minimum correlation coefficient of ≥0.1, within ±500 kb of a given gene, which we reasoned would include the vast majority of putative enhancers[86,145], including those active in only a subset of cells.

We then benchmarked this cMN peak-to-gene set against two alternative scATAC-RNA integrations using subsetted scRNA-seq data from the Mouse Organogenesis Cell Atlas (MOCA)[84]. First we created a neuronal dataset set by integrating our oversampled cMN scATAC profiles with more uniformly sampled sci-RNA neuronal clusters from MOCA (annotated as cholinergic neurons, excitatory neurons, inhibitory neurons, neural progenitor cells, postmitotic premature neurons, primitive erythroid lineage, and stromal cells). We removed any cells that were not collected at e10.5 and e11.5 to age-match our scATAC set. We also performed an scATAC-RNA integration using a more distantly related cell type with minimal sampling overlap, (sci-RNA MOCA Cluster 34 annotated as cardiac muscle lineage) and included non-age-matched cells for this integration. We then generated peak-to-gene links as described above and quantified the total number of links across different RNA integrations.

To quantify and compare the distribution of peak-to-gene links across different genes, we tabulated significant peak-to-gene links ($r > 0.1$ and FDR $< 10^{-4}$) ±50 kb of each gene's TSS. In the case of peaks connected to multiple genes, we selected the link with the lowest FDR value. Next, we generated modified Domain of Regulatory Chromatin (DORC) scores[146] by normalizing all reads in our peak-by-cell matrix by unique fragment count. We then summed these normalized values for all peak-to-gene connections within ±500 kb of each gene TSS for every cell.

## Single-cell multiome (scMultiome)

We performed timed matings, microdissections, dissociation, and FACS to collect GFP-positive cMN3/4, cMN7, cMN12, and sMN cells at e11.5 as described above. Because we did not collect GFP-negative cells at e11.5 or any cells at e10.5, the cellular representation in the scMultiome dataset represents a subset of that in the scATAC-RNA dataset. In addition, instead of generating separate reactions for each cell type, we pooled these cells prior to dissociation, selected GFP-positive cells via FACS, and performed low cell input nuclei isolation (10x Genomics CG000365) and single-cell multiome ATAC + gene expression assay (10x Genomics CG000338) on a total of two pooled replicates. We performed sequencing on a NextSeq 500 for Multiome ATAC and Gene Expression libraries separately, using a custom sequencing recipe for ATAC provided by Illumina. We performed QC, dimensionality reduction, and generated peak-to-gene links as described above using functionality in Signac and ArchR[80,147]. In order to facilitate direct comparison across modalities, we calculated scMultiome fragment depth against our high-confidence scATAC peakset. We calculated multimodal weights for each cell using a weighted nearest neighbor approach[148] and performed ab initio graph-based clustering on our scMultiome cell set. In order to annotate these clusters, we generated cell-cell anchors by defining scMultiome clusters as the query set and our well-annotated scATAC clusters as the reference set. Because each multiome cluster was typically dominated by a single predicted scATAC cluster, we annotated each multiome cluster based on its maximum predicted scATAC membership. As a QC check, we evaluated the concordance of cognate genes and effect sizes of scMultiome versus scATAC-RNA peak-to-gene links, both globally and for established motor neuron markers (Supplementary Data 8, Supplementary Fig. 5, and Fig. 3).

## Single-cell CUT&Tag

We collected cranial motor neurons (GFP-positive cMN3 + cMN4 e11.5, cMN6 e11.5, cMN7 e10.5, and cMN7 e11.5) as described above and performed a modified scCUT&Tag protocol[87]. Briefly, we collected GFP-positive cells directly into fresh antibody buffer (20 mM HEPES pH 7.5, 150 mM NaCl, 0.5 mM spermidine, 1x protease inhibitor (Sigma 11873580001), 2 mM EDTA, 0.05% digitonin, 0.01% NP-40, 1× protease inhibitors and 2% filtered BSA). We centrifuged samples at 450×$g$ for 5 min, washed in 200 uL antibody buffer, centrifuged at 600×$g$ for 3 min, resuspended in 1:50 H3K27Ac primary antibody (monoclonal Rabbit anti-mouse, Abcam ab177178), and incubated overnight at 4 °C with gentle rotation. Nuclei were centrifuged at 600×$g$ for 3 min, washed in 200 uL Dig-Wash-BSA buffer (20 mM HEPES pH 7.5, 150 mM NaCl, 0.5 mM spermidine, 1x protease inhibitor, 0.05% digitonin, 0.01% NP-40, 1x protease inhibitor, and 2% filtered BSA), centrifuged at 600×$g$ for 3 min, resuspended in 1:50 IgG secondary antibody (guinea pig anti-rabbit Novus Biologicals, NBP1-72763), and incubated 1 h at room temperature with gentle rotation. Nuclei were then centrifuged at 600×$g$ for 3 min, washed 3x in Dig300-Wash-BSA (20 mM HEPES pH 7.5, 300 mM NaCl, 0.5 mM spermidine, 1x protease inhibitor, 0.05% digitonin, 0.01% NP-40, 1x protease inhibitors, and 2% filtered BSA), resuspended in 1:20 pAG-Tn5 (EpiCypher 15-1017), and incubated 1 h at room temperature with gentle rotation. Nuclei were centrifuged at 450×$g$ for 3 min, washed 3x in Dig300-Wash-BSA, resuspended in

200 uL tagmentation buffer (20 mM HEPES pH 7.5, 300 mM NaCl, 0.5 mM spermidine, 1x protease inhibitor, 0.05% digitonin, 0.01% NP-40, 1x protease inhibitor, 2% filtered BSA, and 10 mM MgCl$_2$), incubated 1 h at 37 °C with agitation every 15 min. Tagmentation was halted with Stop buffer (20 mM HEPES pH 7.5, 300 mM NaCl, 0.5 mM spermidine, 1x protease inhibitor, 0.05% digitonin, 0.01% NP-40, 1x protease inhibitors, 2% filtered BSA, and 25 mM EDTA), centrifuged at 450×$g$ for 3 min, washed in diluted nuclei buffer (1x ATAC Nuclei Buffer (10x Genomics, PN-2000207) and 2% filtered BSA), centrifuged at 450×$g$ for 3 min, and resuspended in diluted nuclei buffer. Intact nuclei were stained with DAPI and were visualized and counted under fluorescent microscopy. About 70 uL of ATAC master mix (8 μL tagmented nuclei, 7 μL ATAC Buffer B (10x Genomics, PN-2000193), 56.5 μL Barcoding Reagent B (10x Genomics, PN-2000194), 1.5 μL Reducing Agent B (10x Genomics, PN-2000087), 2 μL Barcoding Enzyme (10x Genomics, PN-2000139) was loaded for GEM generation according to the 10x Genomics scATAC v1.1 protocol. Nuclei were diluted if necessary (up to a maximum of 25,000 total nuclei per reaction). Subsequent GEM generation and cleanup steps were performed according to the 10x Genomics scATAC v1.1 protocol. Library prep was also performed using the standard protocol, except that total PCR cycles were increased to 16. All centrifugation steps were performed using a swing-bucket rotor.

## Activity-by-contact (ABC) enhancer predictions

We generated enhancer predictions for four cell types, GFP-positive cMN3 + 4 e11.5, cMN6 e11.5, cMN7 e10.5, and cMN7 at e11.5, adapting the activity-by-contact (ABC) model v0.2[86]. We defined potential enhancer regions by merging scATAC peaksets for each sample. We provided sample-specific H3K27Ac read counts from scCUT&Tag experiments described above. We also provided imputed RNA expression tables for each cell type from the scATAC-scRNA integration described above. We estimated contact frequencies based on the ABC power law function. We evaluated our enhancer predictions against 67 VISTA enhancers classified as positive for cranial nerve, of which 12 had ABC enhancer predictions. Importantly, our ABC predictions also correctly identify the peak and cognate gene for the CREST1 enhancer (VISTA enhancer hs1419), for which both the enhancer locus and cognate gene are known[85].

## Participant whole genome sequencing, reprocessing, SNV/indel calling, and quality control

**Research participants.** Male (49.15%) and female (50.84%) research participants of any age (range 1 month–78 years) diagnosed with congenital cranial dysinnervation disorders and available family members were enrolled in the study. Sex for human participants were collected from clinical information. Reported or genetically inferred sex was used for quality control, relatedness checks, and estimates of contamination. Sex-specific traits were not investigated. This was a cross-sectional observational study of subjects/families enrolled over many decades through dedicated research protocols. Subjects were referred to the research protocol through their physicians, family support groups, or self-referral. They carried a diagnosis of a congenital cranial dysinnervation disorder. There was likely self-selection bias for subjects who have access to health care and/or interest to participate in research but this is not likely to impact results. No compensation was provided to participants.

**SNV/indel calling.** WGS was performed at Baylor Human Genome Sequencing Center through the Gabriella Miller Kids First Pediatric Research Program (dbGaP Study Accession: phs001247). Joint variant calling for all samples was performed at the Broad Institute. We uploaded raw 30X coverage PCR-free WGS data to the Broad Institute's secure Google Cloud server and reprocessed these data through the Broad Institute's production pipeline. We realigned raw read data to

the GRCh38 human reference sequence using BWA-MEM and reprocessed using Broad's Picard Toolkit. We then performed variant calling on the resultant BAM files using the Genome Analysis Toolkit (GATK 4.0 HaplotypeCaller). In the final step of variant calling, we jointly genotyped each site in the genome alongside a collection of over 20,000 reference genomes assembled by the Broad Institute. Joint variant calling provides two crucial advantages over individual or batched genotyping[149]. First, it dramatically improves variant calling accuracy due to (i) clearer distinction between homozygous sites versus missing data; (ii) greater sensitivity to detect rare variants, and (iii) greater specificity against spurious variants. Second, joint calling, by its design, generates a well-calibrated estimate of allele frequency within our cohort against the large gnomAD database. Assuming that the allele frequency of a bona fide Mendelian disease-causing variant is lower than its disease prevalence, this information allows us to exclude variants with implausibly high allele frequencies in population databases[149,150]. Finally, we performed variant filtering using GATK's Variant Quality Score Recalibrator and applied custom hard filters as required.

**SNV/indel QC.** We performed QC at multiple stages of variant calling, performed filtering based on standard sequencing quality metrics (e.g., uniformity of coverage, transition/transversion ratio, indel length profiles), and compared them to our internal database of reference genomes. We used heterozygosity of common variants on chrX and coverage of sites on chrY to confirm reported gender and to identify sex chromosome aneuploidy. We also extracted variant calls from 12,000 well-covered variant sites and used these variants for principal component analysis together with a large reference panel to infer the geographical ancestry of samples, infer pairwise relatedness of the samples, identify unexpected duplicates, and determine cryptic relatedness and unexpected patterns of relatedness within reported families.

### Structural variants
**Variant calling with GATK SV.** We generated an SV callset using the ensemble GATK-SV pipeline (https://github.com/broadinstitute/gatk-sv)[91]. Briefly, we performed joint genotyping and harmonized SV calls from multiple detection tools (Manta, Wham, MELT, GATK-gCNV, and cn.MOPS[151–155]), as well as manual read inspection using IGV[156], and estimated SV allele frequencies against gnomAD SV v2.1[91]. We first excluded any SVs with cohort AF ≥0.005, irrespective of coding or non-coding status. When evaluating de novo and inherited SV candidates, we restricted our callset to 45 and 49 curated pedigrees, respectively. One SV (deletion chr22:27493955-27497536) was identified through manual curation. These SVs were subsequently used for downstream analysis incorporating pedigree non-coding element information.

**Transposable elements.** We also performed a separate bespoke analysis for genome-wide transposon insertions (L1, Alu, and SVA) profiling on the GMKF WGS dataset using xTea[157]. Raw transposon insertions with different features and confidence levels were annotated and processed to generate both rare and de novo insertion lists for further variant interpretation. Beyond basic feature annotations (transposon family, breakpoint, and gene annotations), all insertions were annotated with (1) population allele frequencies (AFs) derived from the 1000 genomes project, gnomAD SV, euL1db, and other polymorphic insertion collections from the literature[91,158–160]; (2) overlapping repeats annotated by RepeatMasker and homopolymers; (3) other gene annotations such as pLI score[161], OMIM disease-causing genes[2], and potential CCDD-related genes. For putative pathogenic rare insertions, we first applied a population AF threshold of 0.01 to remove common polymorphic insertions. We then filtered nested insertions–where a putative insertion landed in an existing insertion from the same transposon family–as they are error-prone in short-read sequencing platforms. Finally, we filtered for all high-confidence

annotations (two_side_tprt_both and two_side_tprt) in affected samples for downstream genetic analysis. For de novo insertions, raw calls of transposon insertions were examined, and only those present in the affected proband but fully absent in both parents (i.e., without a single supporting read) were retained. Trio families with any member bearing an abnormally high number of transposon calls were filtered, as these outlier samples carried excessive noisy signals (clipped and discordant reads), and consequently, false positive calls could affect de novo insertion calling. We then removed insertions that have been reported in populational datasets and known polymorphic insertion collections in the literature. We also filtered out error-prone nested insertions. Finally, high-confidence insertions (feature = two_side_tprt_both) in affected participants were reported as the de novo insertions for further genetic interpretation (Supplementary Data 11).

### Applying cell-type aware filters for human non-coding mutations
**General filtering.** Our original WGS callset contained 49,824,956 variant calls for 899 individuals across 270 distinct families with CCDDs. We loaded these unfiltered variant calls in.vcf format into Hail (https://github.com/hail-is/hail) as a MatrixTable. Multi-allelic variants were split so that all variants are represented in a biallelic format. In splitting multi-allelic variants, spanning deletions were not kept. This resulted in 54,804,014 biallelic variants. These variants were annotated with TOPMed allele frequencies, gnomAD genomes allele frequencies and allele counts, GERP scores, and ClinVar variant pathogenicity labels[92,162–164]. Using native and custom Hail functions, we generated scripts to filter the MatrixTable's variant calls based on custom specifications for variant annotations, variant locus, and call quality filters.

We set the following hard filters for all searches:

gnomAD AF ($<1 \times 10^{-3}$ for dominant/de novo; $<1 \times 10^{-2}$ for recessive)

TOPMed AF ($<1 \times 10^{-3}$ for dominant/de novo; $<1 \times 10^{-2}$ for recessive)

GERP >2

Only return variants that pass all quality filters in the VCF

Genotype quality: >20

Allele balance: >0.15 (heterozygous calls)

**Cell type-specific filtering.** To generate a list of cell type-specific genomic regions of interest for each disease group, we used data from single-cell ATAC-seq experiments performed on mouse cranial motor neurons at e10.5 and e11.5. From here, we implicitly assume that: (i) we have correctly mapped each disease-relevant cell type (at the appropriate timepoint) to its appropriate cognate phenotype; (ii) biologically active cREs are accessible; and (iii) patterns of chromatin accessibility are correlated across species[11,59]. Peaks called on each cMN sample were lifted over from mm10 to hg38, and the converted intervals were concatenated into a single file and overlapping peaks were combined using bedtools *merge*. For disease types with >1 cMN of interest, the master list of intervals for each cranial nerve were again merged using bedtools *merge* to create a list of intervals defining regions accessible in one or both cMNs. This final master list of intervals was used to narrow the total genomic search space for each disease group, with only variants contained in the regions specific to the cMN(s) of interest being retained.

### Modes of inheritance
**SNVs/indels.** To leverage pedigree information, we first stratified our 270 pedigrees into seven major disease categories that shared cell type-specific etiology (CFEOM, FNP, DRS, CFP, Moebius, Ptosis, Ptosis/MGJWS). We further stratified these pedigree groups into subgroups based on four inheritance/phenotype patterns (familial/syndromic; familial/isolated; sporadic (trio)/syndromic; sporadic (trio)/isolated). We incorporated inheritance by only retaining variants that matched

the appropriate mode(s) of inheritance in at least one family per sub-group. For example, for trios, we searched variants obeying de novo, dominant (if either parent was affected), compound heterozygous, and/or homozygous recessive modes of inheritance. For de novo variants, we used Hail's likelihood-based caller (https://github.com/ksamocha/de_novo_scripts). For familial cases, we manually inspected each pedigree structure and specified custom variant searches based on plausible modes of inheritance, including de novo, dominant, compound heterozygous, homozygous recessive, and dominant with incomplete penetrance. In the case of compound heterozygous variant configurations affecting non-coding elements, we defined each scATAC peak as our unit of heredity. Within this framework, one variant in a peak had to be inherited from an unaffected father, and a different variant in the same peak had to be inherited from an unaffected mother. Finally, we performed cohort-level filtering by eliminating any rare candidate variants that were also present in any unaffected individuals in the cohort (for dominant/de novo searches) or that were present in a homozygous state in any unaffected individual (for recessive searches). We excluded candidate variants from one outlier pedigree that failed call QC.

**SVs and transposable elements.** For SV genetic interpretation, we performed inheritance-based searches for dominant/de novo modes of inheritance in the appropriate pedigrees, using the same custom search parameters as described for the SNV/indel framework. We identified all de novo and inherited variants overlapping disease-relevant peaks for each eligible pedigree using the findOverlapPairs() function from the GenomicRanges package.

For TE genetic interpretation, we imported the list of TEs called with xTEA[157] into Hail as a MatrixTable. We performed inheritance-based searches for dominant/de novo modes of inheritance, again using the same custom search parameters as described for the SNV/indel framework. We converted the TE MatrixTable from hg19 coordinates to hg38, and filtered out calls with invalid/unknown contigs, and only included highest confidence calls (Feature info = two_side_tprt_both). We applied estimated gnomAD AF thresholds of 0.01 and 0 for dominant inherited and de novo alleles, respectively. We used the same cell type-specific peak interval/disease group combination described above but added ±15 bp padding to each peak to account for uncertainty in the insertion point.

To identify multi-hit peaks, we aggregated candidate variant results within each cell type/disease pairing by peak and selected any peaks with SNVs/indels and/or SVs present in ≥2 families. For multi-hit tabulation, we excluded any SVs >100 kb or with clear coding etiology. Variants within multi-hit peaks were required to obey the same broad mode of inheritance (i.e., dominant or recessive). In addition, dominant and recessive multi-hit variants could not be present in any unaffected individual across the cohort in the heterozygous and homozygous configuration, respectively. Candidate variants in any previously solved pedigrees were excluded from final tabulation[19,21,22,27,34,98,99,101,103,165–171].

## Permutation testing

To assess the statistical significance of the results that lie within the regions drawn from scATAC sequencing of developing cranial motor neurons, we performed permutation tests to determine whether the regions corresponding to specific cranial motor neurons were enriched for variants. We analyzed dominant inherited and de novo variants separately.

First, we performed a search to find variants using the same thresholds for frequency, conservation, quality, and inheritance, but without limiting the search space to only genomic intervals defined in the scATAC peaks. We then split these results by disease group based on the phenotype of the family to create the genome-wide distribution of candidate variants for each disease group. After examining the

distribution of the number of genome-wide de novo variants per individual after filtering for thresholds, we removed four individuals from the results due to existing significantly outside of the distribution (with the threshold drawn at >75 de novo variants per individual).

We then conducted permutation tests on each disease group, using regioneR[172]. We used the original set of genomic locations from the cranial motor neuron(s) scATAC data to generate a size-matched non-overlapping permuted peak callset. We used the hg38 masked genome from BSGenomes in order to restrict the locations where the randomized peaks could be located. We then counted the number of variants within these regions. This process was repeated for 5000 iterations for each disease group for both de novo and dominant inherited variants.

## Droplet digital PCR (ddPCR) copy number validation

We performed ddPCR droplet generation and droplet reading using the QX200 droplet digital PCR system with Biorad ddPCR Supermix for Probes (Bio-Rad #186-3010). We performed copy number geno-typing for non-coding element hs2757 in pedigrees S190 and S138 using ddPCR Copy Number Assay (Bio-Rad dHsaCNS845311073) and TaqMan Copy Number Reference Assay, human, TERT (Life Tech 4403315) as an internal control. We used the following thermocycler protocol: 1 x [95 °C for 10 min]; 40 x [94 °C for 30 s, 60 °C for 1 min]; 1 x [98 °C for 10 min], 1 x [4 °C hold]. Genotyping was performed in duplicate for all samples.

## Convolutional neural network training and prediction

**Training.** We generated accessibility predictions using *Basenji*[120] after training the network with mouse motor neuron scATAC-seq data. We generated separate predictions for each biological replicate (32 replicates total). To preprocess scATAC-seq data before training the neural network, we first generated bigwigs from the scATAC-seq bam files using mm10 as the reference FASTA. We clipped bigwig coverage at 150 to trim outliers. We generated training, validation, and test sequences with a split of 80% training sequences, 10% validation, and 10% test. We identified regions that should not be included in train-ing sequences with a bed file containing regions that were hard masked in the mm10 fasta file combined with the Encode denylist. The mm10 FASTA file was filtered to only include chromosomes 1–19, X, and Y.

We trained the network retaining the model architecture from the original Basenji manuscript, with seven dilated layers. For this work, the dense output layer contained 32 units (one for each sample). Training was stopped when the correlation coefficient for validation predictions vs. validation experimental data failed to improve after 12 iterations (patience = 12), and the weights from the best iteration were saved as the final model. The complete architecture and list of hyperparameters can be found at https://github.com/arthurlee617/noncoding-mendel under params.json.

**SAD score prediction.** Using this trained network, we generated SNP activity difference (SAD) scores for each human candidate variant by calculating the total difference in predicted reference vs. alternate coverage over a 131,072 bp window centered about each variant site (hg38). Here we made the implicit assumption that a network trained on mouse accessibility data was portable across species within the same cell type[120,173]. We also included four solved CFP pathogenic variants as truth data. For ease of interpretation, we converted all SNV predictions from raw count differences to Z-scores, which fit a normal distribution. To calculate Z-scores for individual candidate indels, we used the SNV-derived scores for our null distribution.

## Non-coding CRISPR mice and binomial ATAC

We performed scATAC-seq for GFP-positive cMN7 e10.5 from two CRISPR-mutagenized mouse lines (*cRE2^Fam4/Fam4* and *cRE2^Fam5/Fam5*)

corresponding to human non-coding pathogenic HCFP1 variants. *cRE2^Fam5/Fam5* corresponds to the pathogenic SNV (mm10 chr6:88224892 A > G) mouse line[34]. *cRE2^Fam4/Fam4* (mm10 chr6:88224893 C > T) was mutated on a C57BL/6J background via CRISPR-Cas9 homology-directed repair at the Boston Children's Hospital Gene Manipulation & Genome Editing Core using guide sequence TAGCAGGTCAACAGGGGCAG and subsequently crossed onto the mixed *Isl^MN*:GFP line described above. For each mutant line, we generated two biological replicates (four replicates total) on embryos from [homozygous mutant × homozygous mutant] timed matings and compared to our wildtype cMN7 e10.5 replicates. For ad hoc comparison across these samples, we performed iterative LSI dimensionality reduction and batch correction using Harmony[174] and normalized coverage by $\log_{10}(nfrags)$. We note that *cRE2^Fam4/Fam4* also harbors an off-target C > T variant 54 bp downstream from the target site (i.e., in addition to the on-target variant). This off-target nucleotide is not mutated in any affected samples. However, we do not explicitly exclude the possibility that this off-target variant contributes to the difference in *cRE2^Fam4/Fam4* accessibility relative to wildtype. For binomial ATAC, we performed [wildtype × homozygous mutant] timed matings for GFP-positive cMN7 from the e10.5 *cRE2^Fam5/Fam5* line, again across two biological replicates.

To test the *cis* effects of the mutant allele on accessibility, we tabulated reference versus mutant allele counts and performed a two-sided exact binomial test:

$$P = \sum_i \Pr(X = i) = \sum_i \binom{n}{i} \pi_0^i (1 - \pi_0)^{n-i} \quad (9)$$
$$i\epsilon\{i : \Pr(X = i) \leq \Pr(X = k)\}$$

where the number of trials, $n$ corresponds to sequencing coverage, the number of successes, $k$ corresponds to reference allele count, and the expected probability of success, $\pi_0$ corresponds to the expected sampling probability of the reference allele under the null hypothesis $H_0$: $\pi = 0.5$.

### Statistics and reproducibility
No statistical method was used to predetermine the sample size. Samples were excluded from the analyses if they failed QC. The experiments were not randomized. The Investigators were not blinded to allocation during experiments and outcome assessment.

### Reporting summary
Further information on research design is available in the Nature Portfolio Reporting Summary linked to this article.

## Data availability
The ATAC-seq, RNA-seq, CUT&Tag, and multiome data generated in this study have been deposited without restriction in the Gene Expression Omnibus database under SuperSeries accession code GSE254090 [https://www.ncbi.nlm.nih.gov/geo/] and is composed of SubSeries GSE254083, GSE254086, GSE254088, GSE254084, GSE254089, and GSE254085. The human WGS data were available through dbGaP authorized access (accession phs001247.v1.p1 [https://www.ncbi.nlm.nih.gov/projects/gap/cgi-bin/study.cgi?study_id=phs001247.v1.p1]; WL is under phs001272.v2.p1 [https://www.ncbi.nlm.nih.gov/projects/gap/cgi-bin/study.cgi?study_id=phs001272.v2.p1]) because they are potentially sensitive, access can be obtained by data access request to the NIH. The in vivo enhancer data generated in this study have been deposited without restriction in the VISTA enhancer database [https://enhancer.lbl.gov] under accession codes as referenced in the manuscript and are listed in Supplementary Table 3 and Supplementary Data 10. Additional processed data are available via Figshare Plus (https://doi.org/10.25452/figshare.plus.26517577.v1)[175] and the UCSC Genome Browser [https://mouse-motor-dev-atac.cells.ucsc.edu]. Source data are provided with this paper.

## Code availability
Custom code to perform analyses from this work is available without restriction at https://github.com/arthurlee617/noncoding-mendel[176].

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

## Acknowledgements

We are indebted to all study participants and their families. We thank Ryosuke Fujiki, Tulsi Patel, Ben Weisburd, Orit Rozenblatt-Rozen, Aviv Regev, Andrew Hill, and Jay Shendure for important technical discus-sions. We thank Max Tischfield, Sarah Izen, Alicia Nugent, Alon Gelber, and Matthew Bauer for technical assistance with bulk and scRNA-seq experiments. Next-generation sequencing for single cell experiments was performed at the Molecular Genetics Core at Boston Children's Hospital. WGS of the CCDD cohort was performed at Baylor College of Medicine through the Gabriella Miller Kids First Pediatric Research Program (dbGaP Study Accession: phs001247). Mouse lines were generated by the Gene Manipulation & Genome Editing Core at Boston Children's Hospital. FACS experiments were performed at the Blavatnik Institute Department of Immunology Flow Cytometry Core Facility at Harvard Medical School, the Boston Children's Hospital Hem/Onc-HSCI Flow Cytometry Research Facility, and the Dana-Farber Flow Cytometry Hematologic Neoplasia and Jimmy Fund Cores at Dana-Farber Cancer Institute.

The work was supported by the Gabriella Miller Kids First Pediatric Research Program NHBLI X01HL132377 (E.C.E.), NEI R01EY027421 (D.G.M., M.E.T., and E.C.E.), NICHD R01HD114353 (L.A.P.), NHGRI R01HG003988 (L.A.P.), NIMH R01MH115957 (M.E.T. and H.B.), DP2-AG072437 (E.A.L.), NINDS K08-NS099502 (M.F.R.), NHLBI T32-HL007627 (M.F.R.), NIGMS T32-GM007748 (M.F.R.), Project ALS A13-0416 (E.C.E.), Boston Children's Hospital - Broad Institute Collaborative Grant (E.C.E.), Boston Children's Hospital Manton Center Rare Disease Fellowships (A.S.L. and B.Z.) and Manton Center Pilot Project Award (B.Z.), Suh Kyungbae Foundation (E.A.L.), the Abramson Fund for Undergraduate Research (C.L.), T32-GM007748 (J.A.J.), 5T32NS007473 (J.A.J.), 5T32EY007145 (J.A.J.), the Harvard Medical School William Randolph Hearst Fund (J.A.J.), and the Boston Children's Hospital Intellectual and Developmental Disabilities Research Center (NIH U54HD090255). The research of M.K. and L.A.P. was conducted at the E.O. Lawrence Berkeley National Laboratory and performed under the US Department of Energy Contract DE-AC02-05CH11231, University of California. E.C.E. is an Investigator of the Howard Hughes Medical Institute.

## Author contributions

A.S.L. and E.C.E. led the experimental design. A.S.L., L.J.A., M.K., W.-M.C., B.P., M.F.R., and A.P.T. performed experiments. A.S.L. led the computa-tional analysis. A.S.L., L.J.A., L.N.F., T.E.C., B.Z., A.S.-J., J.M.F., I.W., X.Z., C.L., K.M.L., M.L., and H.B. performed computational analysis. A.S.L., W.-M.C., B.J.B., V.R., J.A.J., and E.M.E. processed human samples and data. D.G.M., E.A.L., M.E.T., H.B., L.A.P., and E.C.E. provided funding and project super-vision. A.S.L. and E.C.E. wrote the manuscript. A.S.L. devised the study. E.C.E. oversaw the study. All authors read and approved the manuscript.

## Competing interests

D.G.M. is a paid advisor to GlaxoSmithKline, Insitro, and Overtone Therapeutics, and has received research support from AbbVie, Astellas, Biogen, BioMarin, Eisai, Google, Merck, Microsoft, Pfizer, and Sanofi-Genzyme. M.E.T. has received research support and/or reagents from Microsoft, Illumina Inc., Pacific Biosciences, and Ionis Pharmaceuticals. Otherwise, the authors declare that they have no competing interests as defined by Nature Research, or other interests that might be perceived to influence the interpretation of this article.

## Additional information

[1]Department of Neurology, Boston Children's Hospital and Harvard Medical School, Boston, MA, USA. [2]Kirby Neurobiology Center, Boston Children's Hospital, Boston, MA, USA. [3]Manton Center for Orphan Disease Research, Boston Children's Hospital, Boston, MA, USA. [4]Program in Medical and Population Genetics, Broad Institute of MIT and Harvard, Cambridge, MA, USA. [5]Environmental Genomics and Systems Biology Division, Lawrence Berkeley National Laboratory, Berkeley, CA, USA. [6]Howard Hughes Medical Institute, Chevy Chase, MD, USA. [7]Division of Genetics and Genomics, Boston Children's Hospital, Boston, MA, USA. [8]Department of Pediatrics, Harvard Medical School, Boston, MA, USA. [9]Department of Pathology, Boston Children's Hospital, Boston, MA, USA. [10]Department of Pathology, Brigham and Women's Hospital and Harvard Medical School, Boston, MA, USA. [11]Medical Genetics Training Program, Harvard Medical School, Boston, MA, USA. [12]Center for Genomic Medicine, Massachusetts General Hospital, Boston, MA, USA. [13]Department of Neurology, Massachusetts General Hospital and Harvard Medical School, Boston, MA, USA. [14]Harvard College, Cambridge, MA, USA. [15]Centre for Population Genomics, Garvan Institute of Medical Research and UNSW Sydney, Sydney, NSW, Australia. [16]Centre for Population Genomics, Murdoch Children's Research Institute, Melbourne, VIC, Australia. [17]Pediatric Surgical Research Laboratories, Massachusetts General Hospital, Boston, MA, USA. [18]Department of Ophthalmology, Boston Children's Hospital and Harvard Medical School, Boston, MA, USA. ✉e-mail: arthur.lee@childrens.harvard.edu; elizabeth.engle@childrens.harvard.edu

