## [Peer Review File · Nature Communications]

A cell type-aware framework for nominating non-coding variants in Mendelian regulatory disordersReviewers' Comments:

Reviewer #1:

Remarks to the Author:

This study proposed a multi-omic framework integrating chromatin accessibility, histone modification, and gene expression of cranial motor neuron (cMN) development to identify cis-regulatory elements and non-coding variants in congenital cranial dysinnervation disorders (CCDDs). They generated epigenomic profiles of ~86,000 cMNs and related cell types, identifying ~250,000 accessible elements with gene predictions for ~145,000 putative enhancers, ~44 of these elements validated in vivo, suggesting that single cell accessibility is a good predictor of enhancer activity. Reduced variant search space identified known CCDD disease gene regulators MAFB, PHOX2A, CHN1, and EBF3, and new candidates in recurrently mutated enhancers through allelic aggregation. Overall, this work is interesting and provides valuable findings on non-coding variants relevant to CCDDs and proposes a framework to nominate high-impact non-coding variants in Mendelian disorders.

Major concerns:

1. I appreciated the significant amount of time and efforts of authors in working on this project. However, the manuscript is extremely lengthy, with 79 pages and over 1900 lines. It is very difficult to follow because the main text, figures, and figure legends are in separate sections. Particularly, the Methods should be further divided into more subtitles to guide readers. For example, I could not find the methods details supporting Line 184, 208, and many other places.
2. Line 205, 'we performed in vivo humanized enhancer assays on a curated subset (n = 26) of our candidate scATAC peaks...'. It is important to show the starting number of scATAC peaks, and the number of filtered peaks in each steps (absent from the VISTA enhancer database and had peak accessibility/specificity in cMNs and general signatures of enhancer function. The size of peak subset (n = 26) is too small. The authors should provide further evidence to prove that their curation is not arbitrary.
3. Line 355, 'We adopted a gene-centric aggregation approach by first...'. It is unclear what is 'gene-centric aggregation approach' and how to 'connect' non-coding candidate variants to known CCDD disease genes.
4. Line 389, the same concern to the peak-centric aggregation approach. In this case, figure illustration are helpful to readers to understand the structure of 'gene-centric' and 'peak-centric' approach.
5. Line 549, 'Alternatively for common diseases, causal non-coding variants are more abundant, but also confounded by linkage disequilibrium.' Please provide evidence or data to support that CCDD candidate variants were excluded from challenges of linkage disequilibrium (e.g., no highly linked variants in those loci).

Minor concerns:

1. Line 195-196. 'Neither major cluster nor subcluster membership was driven by experimental batch (Extended Data Figure 4d).' It is not clear why the extend fig. 4d could support this claim. Further clarifications were required.
2. Line 531. Please specify 'a search space reduction of 4 orders of magnitude', such as the value before and after prioritization. Moreover, the term 'search space' is vague, is it 'the total number of candidate variants'?
3. Fig. 1a. Human avatar is confusing as the scATAC atlas is based on mouse model.
4. Fig. 1b (iii). Based on the figure, the 'patient WGS variants' is required for 'RNA integration and Cognate gene links'. Is it true?
5. Number the formulas throughout the manuscript.
6. Fig. 7b. The labels 'more closed' and 'more open' are informal.

Reviewer #3:

Remarks to the Author:

In this study, Lee et al developed a framework to prioritize functional/co-segregating non-coding variants utilizing both whole genome sequencing and cell-type-specific putative enhancer information for a group of Mendelian disorders with well-defined cell types of origin. Quality control and validation of single-cell dataset as well as validation of prioritization scheme are thorough with experimental support and re-identification of recently characterized ground-truth variants. The genetic and functional information for the prioritized variants as well as the resource and methodology will be very useful in future research. I recommend the authors to address the following points:

1. As the authors noted, one of the main challenges in using a mouse dataset for assessing the function of human non-coding variants is functional homology between two species, especially when this information could be used for determining pathogenicity. The presented humanized enhancer assays of select scATAC-seq peaks are promising. However, the functional annotation by variant-peak overlap or prediction-based SNP activity difference (SAD) scores might make more sense for human-mouse conserved regions including gene promoter regions (vs. distal enhancer regions). The authors did filtering of human variants based on general conservation score, but would human-mouse conserved TF motifs overlapping with top prioritized candidates shed some light on this issue?
2. The human variant prioritization process is thorough, and the examples with experimental data are illuminating. Since this is a multi-layered process, it would be helpful to have a flow chart or a figure illustrating the process by different approaches and criteria. Also, some examples highlighted

in the text do not have linked display items to refer to, making it challenging to follow the writing:

a. Gene-centric aggregation results for the known CCDD genes (e.g., MAFB, PHOX2A) are not shown in the tables.

b. For the peak-centric results, Table S12 is missing a column for peak membership.

c. Line 404: hs2757 is not shown in any table.

d. It will be informative if Table S13 shows which are the multi-hit genes.

e. EBF3 (line 466) and CHN1 (line 468) examples are not found in any table.

3. Peak-gene linkage is less credible when using external scRNAseq data as opposed to a barcode-matched strategy. Authors provide careful quality control data, but would some of the curious gene matching results (e.g., MN1 example) be resolved in the snATAC/RNA multiome subset of the data? Are most of the prioritized genes (e.g., multi-hit genes) also linked to the variant-overlapping genes in the multiome subset of the data?

4. Compound heterozygosity assignment is based on the assumption that the scATAC peak is a definitive functional unit. Since the mode of inheritance and segregation information is a large part of variant prioritization, this limitation should be noted in the text.

5. Extended Data Figure 7 shows hs1321, while the legend says hs2757.

Reviewer #4:

Remarks to the Author:

Lee et al. presented a pioneering single-cell multi-omic approach to unravel the complex non-coding genetic landscape contributing to congenital cranial dysinnervation disorders (CCDDs) that affect the development of cranial motor neurons (cMN). Meticulous analysis of chromatin accessibility, histone modifications, and gene expression in embryonic mouse cMNs, leading to the identification of approximately 250,000 regulatory elements, marks a significant advancement in our understanding of CCDDs. The application of this comprehensive framework to whole-genome sequences from CCDD pedigrees, enabling a focused search for variant candidates, is particularly commendable. This not only identifies both new and established genes associated with CCDDs, but also introduces a novel pathway for identifying non-coding variants in Mendelian disorders.

The experiments, characterized by thorough execution and extensive quality control, combined single-cell and bulk analyses to ensure the robustness of the dataset. Computational analysis is well documented, enhancing the transparency and reproducibility of the study. The generated cell atlas is a valuable resource for a broader research community, facilitating further discoveries in the field.

The conclusions drawn from this study were supported by the data provided. However, there are a few minor suggestions that could further strengthen this manuscript.

1. This study focused on a narrow developmental window (E10.5 and E11.5) to explore cMN development. To benefit a broader audience and strengthen the rationale for selecting these specific time points, it would be helpful to provide more background information on cMN development. An expanded justification for focusing on these two stages could elucidate the significance of these particular stages in cMN differentiation and maturation.
2. While the study emphasizes that cell type-specific ATAC signals are predictive of cis elements for cognate genes, enhancer validation through transgenic reporter assays focuses on the cis-element's enhancer activity without confirming whether they drive correct gene expression. Assays that test whether the identified cis elements drive specific gene expression in their native cellular context would provide stronger evidence for their functional significance.
3. To enhance the robustness of the inferred ATAC-RNA pairings, a systematic comparison between the CRE-gene linkages determined by multiome analysis and those inferred from separate assays is recommended. Presently, the manuscript provides examples from a few selected loci. A more comprehensive analysis could validate the approach and reliability of inferred linkages across the genome.
4. The comparison mentioned in line 212 might be perceived as misleading because most potential cis-regulatory elements in the VISTA enhancer database were not selected for expression specificity in cMNs. Clarifying this comparison or providing an additional context could help avoid confusion and more accurately reflect the scope and focus of the study.
5. On Line 550, it would be beneficial to specify the number of transgenic mouse lines examined for each enhancer validation experiment. This information adds clarity to the scale and reproducibility of the validation efforts.
6. Supplementary Table 2, including detailed quality control (QC) metrics for each library, such as the number of cells profiled, total reads, and fragment counts, will aid in assessing the data quality and reproducibility of the study.
7. The Methods section appears to contain incorrect citations.

REVIEWER COMMENTS

Reviewer #1 (Remarks to the Author):

This study proposed a multi-omic framework integrating chromatin accessibility, histone modification, and gene expression of cranial motor neuron (cMN) development to identify cis-regulatory elements and non-coding variants in congenital cranial dysinnervation disorders (CCDDs). They generated epigenomic profiles of ~86,000 cMNs and related cell types, identifying ~250,000 accessible elements with gene predictions for ~145,000 putative enhancers, ~44 of these elements validated in vivo, suggesting that single cell accessibility is a good predictor of enhancer activity. Reduced variant search space identified known CCDD disease gene regulators MAFB, PHOX2A, CHN1, and EBF3, and new candidates in recurrently mutated enhancers through allelic aggregation. Overall, this work is interesting and provides valuable findings on non-coding variants relevant to CCDDs and proposes a framework to nominate high-impact non-coding variants in Mendelian disorders.

Major concerns:

1. I appreciated the significant amount of time and efforts of authors in working on this project. However, the manuscript is extremely lengthy, with 79 pages and over 1900 lines. It is very difficult to follow because the main text, figures, and figure legends are in separate sections. Particularly, the Methods should be further divided into more subtitles to guide readers. For example, I could not find the methods details supporting Line 184, 208, and many other places.

We appreciate that a manuscript of this length requires significant time to properly review and we are grateful for the the referee's dedication and recommendations for improvement. In our resubmission we have cut some extraneous details. We have edited and subdivided some of the longer Methods sections as suggested, and we believe that all the methods are included in the main or supplementary text. We have also staged Figures with their matching legends and we anticipate readability will be further improved during the type editing and publication process. That said, because we are aiming for a well documented and detailed manuscript to serve as a guide for investigators across broad disciplines, it remains somewhat long. If there are specific sections that the reviewers or editors would like us to shorten or cut further please let us know.

2. Line 205, 'we performed in vivo humanized enhancer assays on a curated subset (n = 26) of our candidate scATAC peaks....'. It is important to show the starting number of scATAC peaks, and the number of filtered peaks in each steps (absent from the VISTA enhancer database and had peak accessibility/specificity in cMNs and general signatures of enhancer function. The size of peak subset (n = 26) is too small. The authors should provide further evidence to prove that their curation is not arbitrary.

We have added genome-wide peak, VISTA enhancer, specificity, etc. numbers as suggested. We do not intend to compare testing validation rates with overall patterns across the genome, therefore we have also edited the language in the text to avoid potentially misleading comparisons. This now reads:

Therefore, to evaluate the functional conservation of cREs in our cranial motor neuron atlas, we performed *in vivo* humanized enhancer assays on a curated subset (n = 26) of our candidate scATAC peaks (n = 255,804 total) that were absent from the VISTA enhancer database (n = 3,384)⁶⁶ and had peak accessibility/specificity in cMNs (n = 45,813)⁵⁹ and general signatures of enhancer function (i.e., evolutionary conservation and non cMN-specific histone modification data⁶⁷, **Supplementary Table 5, Methods**). Importantly, these peaks and features were not randomly selected and therefore do not necessarily reflect overall patterns across the genome (see refs ^{68,69}).

3. Line 355, 'We adopted a gene-centric aggregation approach by first...'. It is unclear what is 'gene-centric aggregation approach' and how to 'connect' non-coding candidate variants to known CCDD disease genes.

4. Line 389, the same concern to the peak-centric aggregation approach. In this case, figure illustration are helpful to readers to understand the structure of 'gene-centric' and 'peak-centric' approach.

We have added Supplementary Figure 7 to illustrate the peak- and gene-centric approach. The legend reads:

Supplementary Figure 7. Peak- and gene-centric aggregation approaches. Illustration depicting a) peak- and b) gene-centric aggregation approaches. The peak-centric approach aggregates variants that overlapping shared peaks. Conversely, the gene-centric approach aggregates variants overlapping peaks with a shared target gene specified by peak-to-gene links.

5. Line 549, 'Alternatively for common diseases, causal non-coding variants are more abundant, but also confounded by linkage disequilibrium.' Please provide evidence or data to support that CCDD candidate variants were excluded from challenges of linkage disequilibrium (e.g., no highly linked variants in those loci).

Thank you - indeed rare candidate variants could also be confounded by linkage disequilibrium. We have corrected this language in the text, which now reads:

Alternatively for common diseases where causal non-coding variants are more abundant, locus-specific footprinting (in concert with careful demarcation of element boundaries, chromatin accessibility QTL analysis¹²⁶, and statistical fine-mapping¹²⁷) may further resolve causal common variants and identify affected transcription factor binding sites across the genome – all inferred from a single assay.

Minor concerns:

1. Line 195-196. 'Neither major cluster nor subcluster membership was driven by experimental

batch (Extended Data Figure 4d).’ It is not clear why the extend fig. 4d could support this claim. Further clarifications were required.

Thank you, we see how this could require clarification. We have now added an additional paragraph to the Methods that provides needed details. Briefly, we computed all possible pairwise correlations between biological replicates against cluster or subcluster membership. An important detail we now better emphasize in the Methods is that the biological replicates came from different batches, allowing us to compare these replicates directly. In the Figure, we see that biological replicates of the same cell type are most highly correlated. In addition, when further parsing cluster/subcluster membership by replicate, we found that most clusters were not skewed towards one replicate versus the other (‘R1’ vs. ‘R2’), as would otherwise be expected in the presence of pronounced batch effects. This observation was particularly striking at the subcluster level. We found these plots difficult to display appropriately in the manuscript, but we share some screenshots with you here:

Clusters:

Subclusters:

Note ‘neg’ do not have matching biological replicates. Finally, to the extent that Euclidean distance reflects similarity, we observe extremely tight distances among biological replicates in UMAP space (we display an example in Figure 1c). Taken together, we find that cell type of origin is a better predictor of cluster/subcluster membership than replicate/batch.

2. Line 531. Please specify ‘a search space reduction of 4 orders of magnitude’, such as the value before and after prioritization. Moreover, the term ‘search space’ is vague, is it ‘the total number of candidate variants’?

Yes, search space refers to the total number of candidate variants. We have included language in the introduction to further clarify this and have added numerical values as suggested. This now reads:

In Introduction:

We subsequently use this atlas to reduce our candidate variant search space (i.e., total number of eligible variants) and ultimately interpret and nominate non-coding variants among 270 unsolved CCDD pedigrees

In Discussion:

When applying this framework to our CCDD cohort, we achieved a search space reduction of 4 orders of magnitude (i.e., 5.5×10^7 reduced to 5.4×10^3 searchable variants) making non-coding candidate sets human-readable and tractable for functional and mechanistic studies (23.6 candidates per monoallelic pedigree; 13.6 per biallelic pedigree).

3. Fig. 1a. Human avatar is confusing as the scATAC atlas is based on mouse model.

We have now added a mouse avatar to Fig 1a so the conservation between human and mouse can be appreciated.

4. Fig. 1b (iii). Based on the figure, the 'patient WGS variants' is required for 'RNA integration and Cognate gene links'. Is it true?

By themselves RNA integration and gene links can be performed on reference sequence without any variants. In this case the peakset and links would stand alone as a cellular atlas. Conversely, the Mendelian framework related analyses, in particular gene- and peak-centric candidate aggregation do require variants.

5. Number the formulas throughout the manuscript.

Equations 1 through 9 are now numbered.

6. Fig. 7b. The labels 'more closed' and 'more open' are informal.

We have updated the labels in Figure 7 to include the numerical scores.

Reviewer #3 (Remarks to the Author):

In this study, Lee et al developed a framework to prioritize functional/co-segregating non-coding variants utilizing both whole genome sequencing and cell-type-specific putative enhancer information for a group of Mendelian disorders with well-defined cell types of origin. Quality control and validation of single-cell dataset as well as validation of prioritization scheme are thorough with experimental support and re-identification of recently characterized ground-truth variants. The genetic and functional information for the prioritized variants as well as the resource and methodology will be very useful in future research. I recommend the authors to address the following points:

1. As the authors noted, one of the main challenges in using a mouse dataset for assessing the function of human non-coding variants is functional homology between two species, especially when this information could be used for determining pathogenicity. The presented humanized enhancer assays of select scATAC-seq peaks are promising. However, the functional annotation by variant-peak overlap or prediction-based SNP activity difference (SAD) scores might make more sense for human-mouse conserved regions including gene promoter regions (vs. distal enhancer regions). The authors did filtering of human variants based on general conservation score, but would human-mouse conserved TF motifs overlapping with top prioritized candidates shed some light on this issue?

Comparing TF binding sites between human and mouse could provide an additional piece of evidence for human-mouse concordance. As recommended, we annotated the 61 prioritized variants from Supplementary Table 12 with HMR Conserved Transcription Factor Binding Sites track from the UCSC Genome Browser. Because TFBSs can tolerate substantial nucleotide variation, we find many overlapping and closely related motifs across these sites. Of the 61 prioritized variants, we observe discordant TFBS entries in 15. However, the discordant entries are often due to different TFBS annotations from the same family (e.g., PPAR-alpha and PPAR-gamma) or different isoforms of the same gene (e.g., PPAR-gamma1 and PPAR-gamma2), which have nearly identical motifs. We have added these new results to Supplementary Table 12.

2. The human variant prioritization process is thorough, and the examples with experimental data are illuminating. Since this is a multi-layered process, it would be helpful to have a flow chart or a figure illustrating the process by different approaches and criteria. Also, some examples highlighted in the text do not have linked display items to refer to, making it challenging to follow the writing:

All examples in this submission have linked display items. We have generated a flowchart as recommended in new Supplementary Figure 10.

a. Gene-centric aggregation results for the known CCDD genes (e.g., MAFB, PHOX2A) are not shown in the tables.

These results are shown in Table 1 (final page of the manuscript text document) and now refer to it in the main text.

b. For the peak-centric results, Table S12 is missing a column for peak membership.

We have added peak IDs to Table S12.

c. Line 404: hs2757 is not shown in any table.

hs2757 is shown in Table 1. We have now included a reference to Table 1 in the main text.

d. It will be informative if Table S13 shows which are the multi-hit genes.

We have added a column to Supplementary Table 13 listing multi-hit genes.

e. EBF3 (line 466) and CHN1 (line 468) examples are not found in any table.

These results are shown in Table 1.

3. Peak-gene linkage is less credible when using external scRNAseq data as opposed to a barcode-matched strategy. Authors provide careful quality control data, but would some of the curious gene matching results (e.g., MN1 example) be resolved in the snATAC/RNA multiome subset of the data? Are most of the prioritized genes (e.g., multi-hit genes) also linked to the variant-overlapping genes in the multiome subset of the data?

As recommended, we extracted sc-multiome peak-to-gene information for the 21 unique autosomal peaks overlapping prioritized variants from Supplementary Table 12. Of the 21 peaks, 15 have matching cognate gene predictions in the multiome peak-to-gene set.

Importantly, as we elaborate in the manuscript and below, cellular composition of the sc-multiome data represents a subset of the RNA-ATAC data. Therefore, we would not necessarily expect 1:1 correlations across both platforms. We have now further summarized these comparisons in Supplementary Table 12.

4. Compound heterozygosity assignment is based on the assumption that the scATAC peak is a definitive functional unit. Since the mode of inheritance and segregation information is a large part of variant prioritization, this limitation should be noted in the text.

The reviewer has accurately summarized our approach regarding compound heterozygosity. We have added language to the main text to better emphasize this point. This now reads:

We identified 5,353 unique segregating SNVs/indels (3,163 *de novo*/dominant, 1,173 homozygous recessive, and 1,017 compound heterozygous) that overlapped cMN-relevant peaks of accessible chromatin (23.6 and 13.6 candidates per monoallelic and

biallelic pedigree, respectively). We only considered compound heterozygous variants observed in the same peak.

5. Extended Data Figure 7 shows hs1321, while the legend says hs2757.

We apologize for this typo and have corrected the legend in new Supplementary Figure 8.

Reviewer #4 (Remarks to the Author):

Lee et al. presented a pioneering single-cell multi-omic approach to unravel the complex non-coding genetic landscape contributing to congenital cranial dysinnervation disorders (CCDDs) that affect the development of cranial motor neurons (cMN). Meticulous analysis of chromatin accessibility, histone modifications, and gene expression in embryonic mouse cMNs, leading to the identification of approximately 250,000 regulatory elements, marks a significant advancement in our understanding of CCDDs. The application of this comprehensive framework to whole-genome sequences from CCDD pedigrees, enabling a focused search for variant candidates, is particularly commendable. This not only identifies both new and established genes associated with CCDDs, but also introduces a novel pathway for identifying non-coding variants in Mendelian disorders.

The experiments, characterized by thorough execution and extensive quality control, combined single-cell and bulk analyses to ensure the robustness of the dataset. Computational analysis is well documented, enhancing the transparency and reproducibility of the study. The generated cell atlas is a valuable resource for a broader research community, facilitating further discoveries in the field.

The conclusions drawn from this study were supported by the data provided. However, there are a few minor suggestions that could further strengthen this manuscript.

1. This study focused on a narrow developmental window (E10.5 and E11.5) to explore cMN development. To benefit a broader audience and strengthen the rationale for selecting these specific time points, it would be helpful to provide more background information on cMN development. An expanded justification for focusing on these two stages could elucidate the significance of these particular stages in cMN differentiation and maturation.

Thank you, we appreciate the value of adding more background for the non-specialist. We have elaborated on cMN development in the context of our sample choices in the introductory paragraph of the results. Briefly, we prioritized the highest yield timepoints required for a tractable number of experiments and analysis. Based on our knowledge of CCDD coding mutations, we reasoned that the major known modes of pathogenicity disrupted motor nucleus birth/development/maintenance and/or subsequent axon guidance, which we have previously modelled at e10.5 and e11.5. In addition, we have cited additional references showing that e10.5 and e11.5 are timepoints that overlap known motor nucleus birth, as well as axon extension into the periphery. Finally, we have

mentioned the limitations of restricting to these timepoints in the Discussion—namely that any mutations that disrupt a process outside of nucleus birth or axon guidance during these timepoints may not be represented in our dataset. This now reads:

In introduction:

cMN birth and development occur continuously over a period of weeks in early human embryos and days in mice^{34,37}. More specifically, birthdating studies show that cMN3, 4, and 7 overlap mouse/mouse-equivalent stages e9.25 through e12.0^{34,38–41}. In addition, their axons have extended into the periphery and are forming nerve branches at time points overlapping e10.5 and e11.5^{42–47}. Finally, for the known CCDD genes, mRNA expression and/or observed cellular defects typically overlap key developmental timepoints e10.5 and e11.5 in mice – both for cellular identity-related transcription factor^{42,48–51} and axon guidance-related^{22,52,53} variants. Therefore, we captured e10.5 and e11.5 embryonic timepoints for each cMN sample, reasoning that a major proportion of both relevant cellular birth and initial axonal growth and guidance would be represented at one or both of these ages^{34,37}.

In discussion:

Moreover, we sampled cMNs at e10.5 and e11.5 based on developmental patterns of previously described protein-coding mutations, but we do not exclude the possibility that novel disease mutations may also be relevant at different timepoints. Therefore, while our genetic framework can generalize to other disorders, we suspect that appropriate prospective or retrospective epigenomic cell sampling will benefit from highly detailed biological knowledge of each specific disease process.

2. While the study emphasizes that cell type-specific ATAC signals are predictive of cis elements for cognate genes, enhancer validation through transgenic reporter assays focuses on the cis-element's enhancer activity without confirming whether they drive correct gene expression. Assays that test whether the identified cis elements drive specific gene expression in their native cellular context would provide stronger evidence for their functional significance.

Comparing *in situ* enhancer versus gene expression assays provides correlative but valuable information regarding the role of a given cis regulatory element. Because systematic *in situ* comparisons across the genome is beyond the scope of this manuscript, we have carefully curated existing literature where available, added references to the text where appropriate, and present evidence below:

i) **hs1419 and hs1321 (Isl1 cognate gene Supplementary Figure 6, this work)**

Isl1 gene expression - image from Zhuang et al <https://doi.org/10.1016/j.ggp.2013.07.001>

ii) **hs2678 (Phox2a cognate gene Figure 3, this work)**

Phox2a gene expression - Images from Tiveron et al.
<https://doi.org/10.1523/JNEUROSCI.16-23-07649> and Gaufo et al.
<https://doi.org/10.1242/dev.127.24.5343>

iii) Cluster C19 (cMN4) *Rgs4* gene score (Supplementary Figure 2, this work)

Rgs4 cMN4 gene expression - images from Grillet et al.
<https://doi.org/10.1523/JNEUROSCI.23-33-10613.2003>

3. To enhance the robustness of the inferred ATAC-RNA pairings, a systematic comparison between the CRE-gene linkages determined by multiome analysis and those inferred from

separate assays is recommended. Presently, the manuscript provides examples from a few selected loci. A more comprehensive analysis could validate the approach and reliability of inferred linkages across the genome.

We have modified new Supplementary Figure 5 to incorporate the distribution of peak-to-gene effect sizes (i.e., per-link correlation coefficient) for the ATAC-RNA versus multiome platforms across the entire genome. The two assays are significantly correlated. Interestingly we also observe that multiome peak-to-gene links show lower global estimated effect sizes when compared to ATAC-RNA. This phenomenon can also be observed in Figure 3g. There are myriad potential reasons for this observation. Most obvious is that the collection strategies were slightly different for each assay due to more demanding cell input requirements for scMultiome (described in Methods). This means that the sampled cellular composition of the multiome assay is a subset of the ATAC-RNA assays (i.e., e11.5 motor neurons), which may contribute to these differences. Because this phenomenon could be of interest to other investigators choosing between RNA-ATAC and multiome, we have added further description to the Methods. We thank you for the comment. This now reads:

We performed timed matings, microdissections, dissociation, and FACS to collect GFP-positive cMN3/4, cMN7, cMN12, and sMN cells at e11.5 as described above. Because we did not collect GFP-negative cells at e11.5 or any cells at e10.5, the cellular representation in the scMultiome dataset represents a subset of that in the scATAC-RNA dataset. In addition, instead of generating separate reactions for each cell type, we pooled these cells prior to dissociation, selected GFP-positive cells via FACS, and performed Low Cell Input Nuclei Isolation (10x Genomics CG000365) and Single Cell Multiome ATAC + Gene Expression assay (10x Genomics CG000338) on a total of two pooled replicates. We performed sequencing on a NextSeq 500 for Multiome ATAC and Gene Expression libraries separately, using a custom sequencing recipe for ATAC provided by Illumina. We performed QC, dimensionality reduction, and generated peak-to-gene links as described above using functionality in Signac and ArchR^{80,146}. In order to facilitate direct comparison across modalities, we calculated scMultiome fragment depth against our high confidence scATAC peakset. We calculated multimodal weights for each cell using a weighted nearest neighbour approach¹⁴⁷ and performed *ab initio* graph-based clustering on our scMultiome cell set. In order to annotate these clusters, we generated cell-cell anchors by defining scMultiome clusters as the query set and our well-annotated scATAC clusters as the reference set. Because each multiome cluster was typically dominated by a single predicted scATAC cluster, we annotated each multiome cluster based on its maximum predicted scATAC membership. As a QC check, we evaluated concordance of cognate genes and effect sizes of scMultiome versus scATAC-RNA peak-to-gene links, both

globally and for established motor neuron markers (**Supplementary Table 12, Supplementary Figure 5, and Figure 3**).

4. The comparison mentioned in line 212 might be perceived as misleading because most potential cis-regulatory elements in the VISTA enhancer database were not selected for expression specificity in cMNs. Clarifying this comparison or providing an additional context could help avoid confusion and more accurately reflect the scope and focus of the study.

The reviewer is correct that we do not intend to imply equivalent selection of VISTA enhancers and cMN scATAC enhancers. We have edited this paragraph to avoid any potentially misleading comparisons. It now reads:

Therefore, to evaluate the functional conservation of cREs in our cranial motor neuron atlas, we performed *in vivo* humanized enhancer assays on a curated subset ($n = 26$) of our candidate scATAC peaks ($n = 255,804$ total) that were absent from the VISTA enhancer database ($n = 3,384$)⁶⁶ and had peak accessibility/specificity in cMNs ($n = 45,813$)⁵⁹ and general signatures of enhancer function (i.e., evolutionary conservation and non cMN-specific histone modification data⁶⁷, **Supplementary Table 5, Methods**). Importantly, these peaks and features were not randomly selected and therefore do not necessarily reflect overall patterns across the genome (see refs ^{68,69}). We detected positive enhancer activity (any reporter expression) in 65% (17/26) of candidates. Moreover, 11 of the 17 validated enhancers (65%, 42% overall) recapitulate the anatomic expression patterns (motor neuron expression) predicted from the scATAC accessibility profiles to the resolution of individual nuclei/nerves. For these curated examples, we find that high quality single cell accessibility profiles are highly predictive of cell type specific-regulatory activity.

5. On Line 550, it would be beneficial to specify the number of transgenic mouse lines examined for each enhancer validation experiment. This information adds clarity to the scale and reproducibility of the validation efforts.

We have added numbers and sample sizes for all wildtype and mutant reporter constructs where applicable in Supplementary Table 5 and 14, Results, Figure 3 and 6, and Supplementary Figure 8 and 11.

6. Supplementary Table 2, including detailed quality control (QC) metrics for each library, such as the number of cells profiled, total reads, and fragment counts, will aid in assessing the data quality and reproducibility of the study.

Yes, we agree. We have updated Supplementary Table 2 to include nFragments, FRIP, Reads in TSS, and several other QC metrics, thank you.

7. The Methods section appears to contain incorrect citations.

We apologize for this formatting issue. We have corrected these citations.

Reviewers' Comments:

Reviewer #1:

Remarks to the Author:

The authors have addressed all my concerns. The manuscript has substantially improved. Overall, this is a valuable study to the field.

Reviewer #3:

Remarks to the Author:

The authors have satisfactorily addressed all the comments raised by me.

Reviewer #4:

Remarks to the Author:

The authors have satisfactorily addressed my previous concerns.

Reviewer #1 (Remarks to the Author):

The authors have addressed all my concerns. The manuscript has substantially improved. Overall, this is a valuable study to the field.

Thank you.

Reviewer #3 (Remarks to the Author):

The authors have satisfactorily addressed all the comments raised by me.

Thank you.

Reviewer #4 (Remarks to the Author):

The authors have satisfactorily addressed my previous concerns.

Thank you.